# Mechanistic insights into the structure-based design of a CspZ-targeting Lyme disease vaccine

Kalvis Brangulis [1,2] ✉, Jill Malfetano[3], Ashley L. Marcinkiewicz[3,4], Alan Wang [3,11], Yi-Lin Chen[5,6], Jungsoon Lee[5,6], Zhuyun Liu[5,6], Xiuli Yang[7], Ulrich Strych [5,6], Dagnija Tupina[1], Inara Akopjana[1], Maria-Elena Bottazzi [5,6,8], Utpal Pal [7], Ching-Lin Hsieh[9] ✉, Wen-Hsiang Chen [5,6] ✉ & Yi-Pin Lin [3,4,10] ✉

*Borrelia burgdorferi* (*Bb*) causes Lyme disease (LD), one of the most common vector-borne diseases in the Northern Hemisphere. Here, we solve the crystal structure of a mutated *Bb* vaccine antigen, CspZ-YA that lacks the ability to bind to host complement factor H (FH). We generate point mutants of CspZ-YA and identify CspZ-YA$_{I183Y}$ and CspZ-YA$_{C187S}$ to trigger more robust bactericidal responses. Compared to CspZ-YA, these CspZ-YA mutants require a lower immunization frequency to protect mice from LD-associated inflammation and bacterial colonization. Antigenicity of wild-type and mutant CspZ-YA proteins are similar, as measured using sera from infected people or immunized female mice. Structural comparison of CspZ-YA with CspZ-YA$_{I183Y}$ and CspZ-YA$_{C187S}$ shows enhanced interactions of two helices adjacent to the FH-binding sites in the mutants, consistent with their elevated thermostability. In line with these findings, protective CspZ-YA monoclonal antibodies show increased binding to CspZ-YA at a physiological temperature (37 °C). In summary, this proof-of-concept study applies structural vaccinology to enhance intramolecular interactions for the long-term stability of a *Bb* antigen while maintaining its protective epitopes, thus promoting LD vaccine development.

Active immunization aims to trigger host immunity to eliminate pathogens during acute and subsequent infections[1]. Vaccination needs to be safe, and the number of immunizations should be limited, while still providing protection[2,3]. However, some surface antigens from infectious agents, even in their recombinant forms, are less immunogenic, resulting in inefficient pathogen elimination[4]. That challenge facilitates the development of multiple antigen engineering strategies to enhance antigenicity[5], one of which is the structure-based vaccine design[6,7]. This strategy examines the structures of wild-type and mutant proteins seeking to improve protein stability and immunogenicity[6,7]. Structure-based vaccine design has demonstrated its suitability for antigen engineering against numerous pathogens to

[1]Latvian Biomedical Research and Study Centre, Riga, Latvia. [2]Department of Human Physiology and Biochemistry, Riga Stradins University, Riga, Latvia. [3]Division of Infectious Diseases, Wadsworth Center, NYSDOH, Albany, NY, USA. [4]Department of Infectious Disease and Global Health, Cummings School of Veterinary Medicine, Tufts University, North Grafton, MA, USA. [5]Department of Pediatrics, National School of Tropical Medicine, Baylor College of Medicine, Houston, TX, USA. [6]Texas Children's Hospital Center for Vaccine Development, Houston, TX, USA. [7]Department of Veterinary Medicine, Virginia-Maryland Regional College of Veterinary Medicine, University of Maryland, College Park, MD, USA. [8]Department of Biology, Baylor University, Waco, TX, USA. [9]Department of Molecular Biosciences, The University of Texas at Austin, Austin, TX 78712, USA. [10]Department of Biomedical Sciences, SUNY Albany, Albany, NY, USA. [11]Present address: Pomona College, Claremont, CA, USA. ✉e-mail: kalvis@biomed.lu.lv; karstyoyo@gmail.com; Wen-Hsiang.Chen@bcm.edu; Yi-Pin.Lin@tufts.edu

prevent infectious diseases, including the most recent outbreak of COVID-19[8–10] (for a review, see ref. [11]). Nonetheless, the molecular mechanisms underlying the association between antigen stability and robust immunogenicity are still under investigation.

Lyme disease, also known as Lyme borreliosis, is the most common vector-borne disease in many parts of the Northern Hemisphere, with the number of human cases continuously rising (~476,000 cases in the United States reported in 2022), but no effective human vaccines are commercially available[12,13]. As causative agents, multiple species of the spirochete bacteria, *Borrelia burgdorferi* sensu lato (also known as *Borreliella burgdorferi* or Lyme borreliae) are carried by infected *Ixodes* ticks and migrate to vertebrate hosts through tick bites[14]. Amongst those Lyme borreliae species, *B. burgdorferi* sensu stricto (hereafter *B. burgdorferi*) is the most prevalent human infectious Lyme borreliae species in North America, while other human infectious species (e.g., *B. afzelii*, *B. garinii*, and *B. bavariensis*) are prevalent in Eurasia[15]. Upon introduction to hosts, Lyme borreliae colonize the tick bite sites in the skin and then disseminate through the bloodstream to distal organs, causing arthritis, carditis, and/or neurological symptoms (i.e., neuroborreliosis)[16]. A human Lyme disease vaccine (LYMErix) was commercialized 20 years ago but then withdrawn from the market (detailed in refs. [17,18]). A second-generation vaccine is in clinical trials[19–22]. These vaccines target a Lyme borreliae protein, OspA, that is produced abundantly when bacteria are in ticks. However, OspA production is reduced after bacteria enter mammalian hosts[23,24]. Therefore, repeated boosters are required for OspA-targeting vaccines to maintain protective levels of antibodies, challenging Lyme disease vaccine development[18,25]. Efforts thus were taken to revise vaccine regimens of OspA to overcome such hurdles and recently have seen some advancement in enhancing the duration of protective antibodies present in the hosts[26,27].

A second approach to avoiding frequent immunizations is to explore alternative vaccine candidates. Lyme borreliae produce other outer surface proteins, including CspZ (also known as BbCRASP-2[28,29]). CspZ promotes bacterial dissemination to distal tissues by evading the complement system, the first-line innate immune defense in vertebrates in the blood, through binding and recruiting a host complement inhibitor, factor H (FH)[28]. Although CspZ is not found in every Lyme borreliae strain[30], serologically confirmed and/or symptomatic human Lyme disease patients in North America and Eurasia, all develop elevated levels of antibodies that recognize CspZ[31,32]. These findings suggest the production of CspZ in most human infectious Lyme borreliae strains or species. Additionally, CspZ is only produced after Lyme borreliae invade vertebrate hosts and, as such, induces CspZ-specific immune responses after natural infection[33,34]. These observations underscore the potential of employing this protein as an attractive Lyme disease vaccine candidate. However, in mice, vaccination with the wild-type CspZ protein formulated with Freund's adjuvant or aluminum hydroxide did not protect from Lyme borreliae colonization and Lyme disease-associated manifestations. One possibility is that CspZ's protective epitopes are saturated by FH, which would not allow this protein to induce sufficient bactericidal antibodies to efficiently eliminate bacteria in vivo. We thus generated a CspZ-Y207A/Y211A mutant (CspZ-YA) that was shown to be deficient in FH-binding[35], leading to the exposure of the epitopes present in the region of CspZ-YA that is bound by FH[36,37]. We demonstrated the protectivity of TiterMax Gold-adjuvanted CspZ-YA against tickborne infection of multiple human infectious Lyme borreliae strains and species and correlated this with CspZ-YA-induced antibodies that uniquely recognize the epitopes surrounding the FH-binding site[37,38]. These results and the availability of the three-dimensional structure allow CspZ-YA to be used for structure-based vaccine design to ultimately improve efficacy.

In this study, we identified and tested CspZ-YA mutants that showed improved capability to induce bacterial killing and prevent Lyme disease-associated manifestations. We then examined the three-dimensional structures and long-term stability of these identified CspZ-YA mutant proteins at 37 °C to explore the possibility of efficacy enhancement.

## Results

### Structure-based vaccine design identified the amino acid residues enhancing the stability of CspZ-YA

To elucidate the structural basis for the efficacy of CspZ-YA (Y207A/Y211A mutant deficient in FH-binding), we crystallized the recombinant untagged version of CspZ-YA and obtained the structure at 1.90 Å resolution (PDB ID 9F1V) (Fig. 1A and Table S1). Using the CCP4MG software[39], we superimposed CspZ-YA with the previously determined crystal structures of *B. burgdorferi* B31 CspZ from the CspZ/SCR6-7 complex (PDB ID 9F7I; RMSD 0.9 Å) and the CspZ/SCR7 complex (PDB ID 6ATG; RMSD 0.74 Å)[40]. Similar to CspZ, CspZ-YA is an all-α-helical protein composed of nine α-helices (denoted as helices A to I) (Fig. 1A). The superimposed structures revealed that the Y211A mutation extended helix I by three residues (A211, K212, K213) (inlet figures in Fig. 1A). This extension of helix I and the Y207A mutation resulted in an altered conformation of the loop between helixes H and I (loop H/I, inlet figures in Fig. 1A). Such an orientation prevented the formation of a salt bridge between the guanidine group of R206 (located on loop H/I) and the carboxylate of E186 that potentially has a stabilizing effect on the local conformation (inlet figures of Fig. 1A). Additionally, when comparing the distance between R206 and E186 in CspZ and CspZ-YA using Molecular Dynamic (MD) simulation, we found this distance in CspZ fluctuating between 2.5 and 22.2 Å, but frequently within the range of salt-bridge formation (≤4 Å). However, in CspZ-YA, the distance fluctuates between 2.6 and 31.8 Å but mostly greater than 4 Å. Such a distance is unfavorable for the formation of the salt bridge, suggesting the destabilization of CspZ-YA (Fig. S1). Overall, the availability of this high-resolution structure provides a foundation for further structure-based vaccine design of CspZ-YA[35].

We then designed CspZ-YA variants to enhance the stability of CspZ-YA by mutating four categories of amino acids[6,7]: (1) prolines at loop regions to decrease folding entropy (i.e., T67P, F105P), (2) polar residues to reduce surface hydrophobicity (i.e., I80T, I115T), (3) bulky hydrophobic residues to fill internal cavities, (i.e., V142M, I183Y, and G193M), and (4) charge repulsions to disrupt FH binding (i.e., K136E) (Fig. 1B). The recombinant versions of CspZ-YA with T67P, I80T, F105P, I115T, K136E, V142M, I183Y, and G193M were produced in *E. coli* with histidine tags. Additionally, to avoid the formation of intermolecular disulfide bonds, resulting in the risk of protein aggregations[41,42], we substituted two cysteine residues, C53S and C187S, to generate untagged CspZ-YA$_{C53S}$ and CspZ-YA$_{C187S}$ (Fig. 1B). The histidine-tagged and untagged proteins of CspZ-YA were also produced as controls. We found that CspZ-YA$_{C53S}$ was aggregated and insoluble (data not shown) while other CspZ-YA mutants were soluble and did not show differences in their secondary structures by circular dichroism, compared to CspZ-YA (Fig. S2). Therefore, all variants except CspZ-YA$_{C53S}$ were moved forward to the following studies.

### Immunization with CspZ-YA$_{I183Y}$ and CspZ-YA$_{C187S}$ of pre-adolescent C3H-HeN mice elicited robust borreliacidal antibody titers after two doses

To characterize the impact of these mutagenized amino acid residues on immunogenicity, we immunized mice with each of these CspZ-YA mutant proteins or CspZ-YA with different frequencies (number of vaccinations). The titers of anti-CspZ IgG in the sera at fourteen days post-last immunization (14 dpli) were determined (Fig. 2A). In any case, mice inoculated with any CspZ-YA proteins mounted significantly higher titers of CspZ IgG than those from PBS-inoculated control mice (Fig. S3). The IgG titers increased as the immunization frequency increased (Fig. S3). No significantly different titers were observed between the groups (Fig. S3A–C). These results indicate that the mutagenesis of these amino acid residues did not affect the overall IgG

 

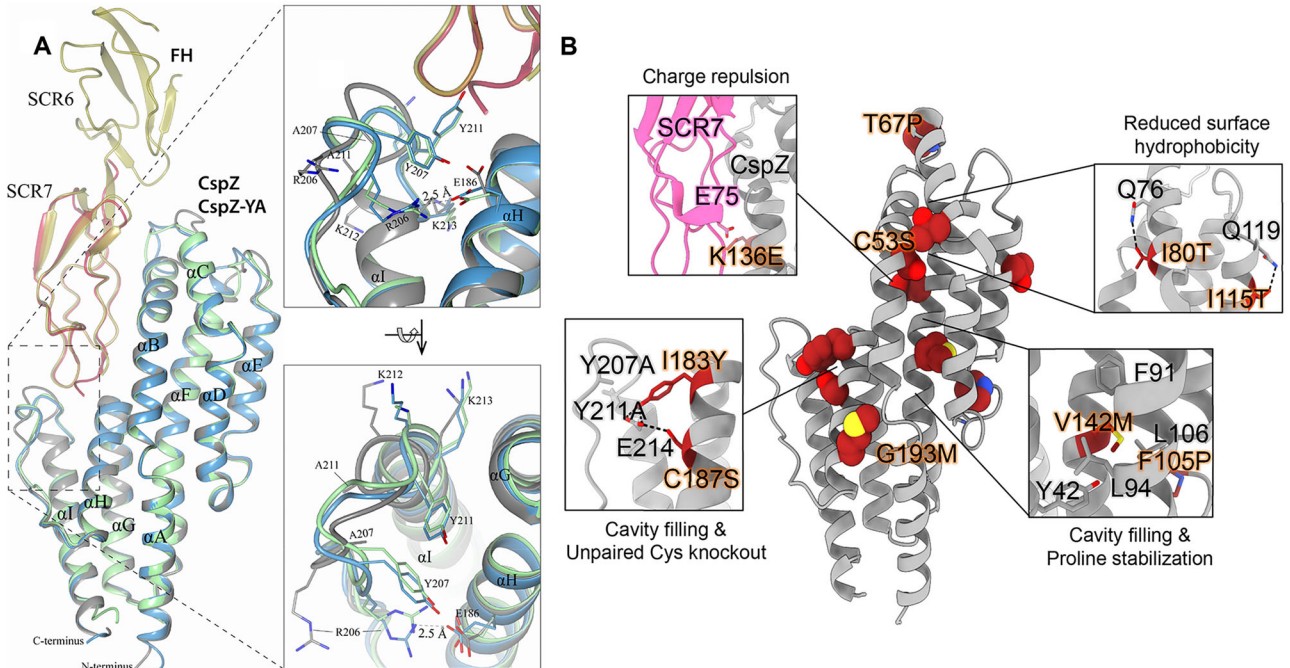

**Fig. 1 | The X-ray structure of CspZ-YA and the mutagenesis of amino acid residues in CspZ-YA by structure-based vaccine design. A** The crystal structure of CspZ-YA (gray; PDB ID 9F1V) is superimposed with the structure of *B. burgdorferi* B31 CspZ (blue) from the CspZ/SCR6-7 complex (PDB ID 9F7I; RMSD 0.9 Å) and *B. burgdorferi* B31 CspZ (green) from the CspZ/SCR7 (red) complex (PDB ID 6ATG; RMSD 0.74 Å). A portion of the electron density map from the newly solved structure of CspZ-YA is shown in Fig. S8A. All nine α-helices of CspZ are labeled (αA-αI). The inlet figures show the loop region between helices H and I, highlighting residues Y207 and Y211 in CspZ from *B. burgdorferi* strain B31 and the mutated residues A207 and A211 in CspZ-YA. Residues K212 and K213 found in the loop region in CspZ and in the extended helix I in CspZ-YA are shown. The interaction between residues R206 and E186 in CspZ is further indicated. The structure is presented from top and side views. **B** Design landscape of CspZ-YA (PDB ID 9F1V) is shown as a ribbon diagram with the side chains of the mutated amino acid residues shown as spheres. Insets highlight the position and side chains of selected stabilizing mutations. Side chains in each inset are shown as dark red sticks with sulfur atoms in yellow, nitrogen atoms in blue, and oxygen atoms in red.

titers after vaccination. We then examined the ability of sera from 14 dpli to kill Lyme borreliae in vitro using *B. burgdorferi* strain B31-A3 as a model as this strain belongs to genotype *ospC* type A that is most prevalent in North America[43]. Different dilutions of the sera were incubated with *B. burgdorferi* and the percent motile bacteria after incubation was determined to extrapolate $BA_{50}$ values, the dilution rate of the sera that kills 50% of bacteria. When we compared these sera's $BA_{50}$ values, those values from any CspZ-YA proteins or variants increased when the immunizations were more frequent (Fig. 2B–G and Table S2). The CspZ-YA mutants displayed no significantly different $BA_{50}$ values from their parental CspZ-YA proteins after immunization once (Fig. 2C). Remarkably, with just two immunizations, CspZ-YA$_{I183Y}$ and CspZ-YA$_{C187S}$ triggered 3.49- and 4.89-fold greater levels of $BA_{50}$ values than the parental CspZ-YA, respectively (Fig. 2E and Table S2). After three immunizations, compared to CspZ-YA, CspZ-YA$_{I183Y}$, and CspZ-YA$_{C187S}$ induced 5.08- and 3.6-fold higher BA50 values, respectively (Fig. 2G and Table S2).

**The I183Y and C187S mutations allowed CspZ-YA to protect pre-adolescent C3H-HeN mice from *B. burgdorferi* infection with fewer immunizations**

We aimed to determine the ability of I183Y and C187S to reduce the protective immunization frequency of CspZ-YA against Lyme disease infection. Mice were immunized with CspZ-YA or variant proteins using different immunization frequencies, followed by infection using ticks carrying *B. burgdorferi* B31-A3 (Fig. 2A). We also included two control groups, PBS and lipidated OspA. At 42 dpli, we measured the bacterial burdens in different tissues and replete nymphs and detected the levels of IgG against C6 peptide, the commonly used Lyme disease serodiagnostic target (Figs. 2A, 3)[44].

We found that the fed nymphs from the mice inoculated with PBS, OspA, CspZ-YA, or CspZ-YA$_{C187S}$ under any immunization frequency accounted for similar levels of bacterial burden (Fig. 3A, G, M). This is in agreement with prior findings that vaccinating three times with OspA or CspZ-YA does not eliminate *B. burgdorferi* in fed ticks[37,45]. Mice immunized once with any tested antigens were all seropositive (Fig. 3B) and had significantly greater bacterial loads in indicated tissues than uninfected mice (Fig. 3C–F). In contrast, mice inoculated three times with the test antigens were seronegative for C6 IgG (Fig. 3N) with indistinguishable levels of bacterial burdens from uninfected mice (Fig. 3O–R). All mice inoculated twice with PBS, CspZ-YA (untagged or histidine-tagged), and four out of five mice immunized twice with OspA were seropositive for C6 IgG and had significantly higher levels of bacterial loads than uninfected mice (Fig. 3H–L). However, CspZ-YA$_{C187S}$- or CspZ-YA$_{C187S}$- inoculated mice showed no significantly different levels of bacterial burdens at tissues or seropositivity, compared to uninfected mice (Fig. 3H–L). These results identified that two doses of CspZ-YA$_{C187S}$ or CspZ-YA$_{I183Y}$ but not CspZ-YA and OspA prevented colonization with *B. burgdorferi* and Lyme disease seroconversion. We further determined the severity of *B. burgdorferi*-triggered inflammation at joints in mice after two immunizations by histologically examining the mouse ankles. At 21 dpli, in CspZ-YA- (untagged or histidine-tagged) or PBS-inoculated mice, we found an elevated number of mononuclear cells infiltrating the tendon, connective tissues, and muscles (arrows in Fig. 4). One out of five OspA-immunized mice was protected from such *B. burgdorferi*-induced joint inflammation, while the rest four mice in this vaccination group developed such an inflammation. These mice had indistinguishable scores of the inflammation from PBS-inoculated, *B. burgdorferi*-infected mice (Inlet figure in Fig. 4). However, similar to

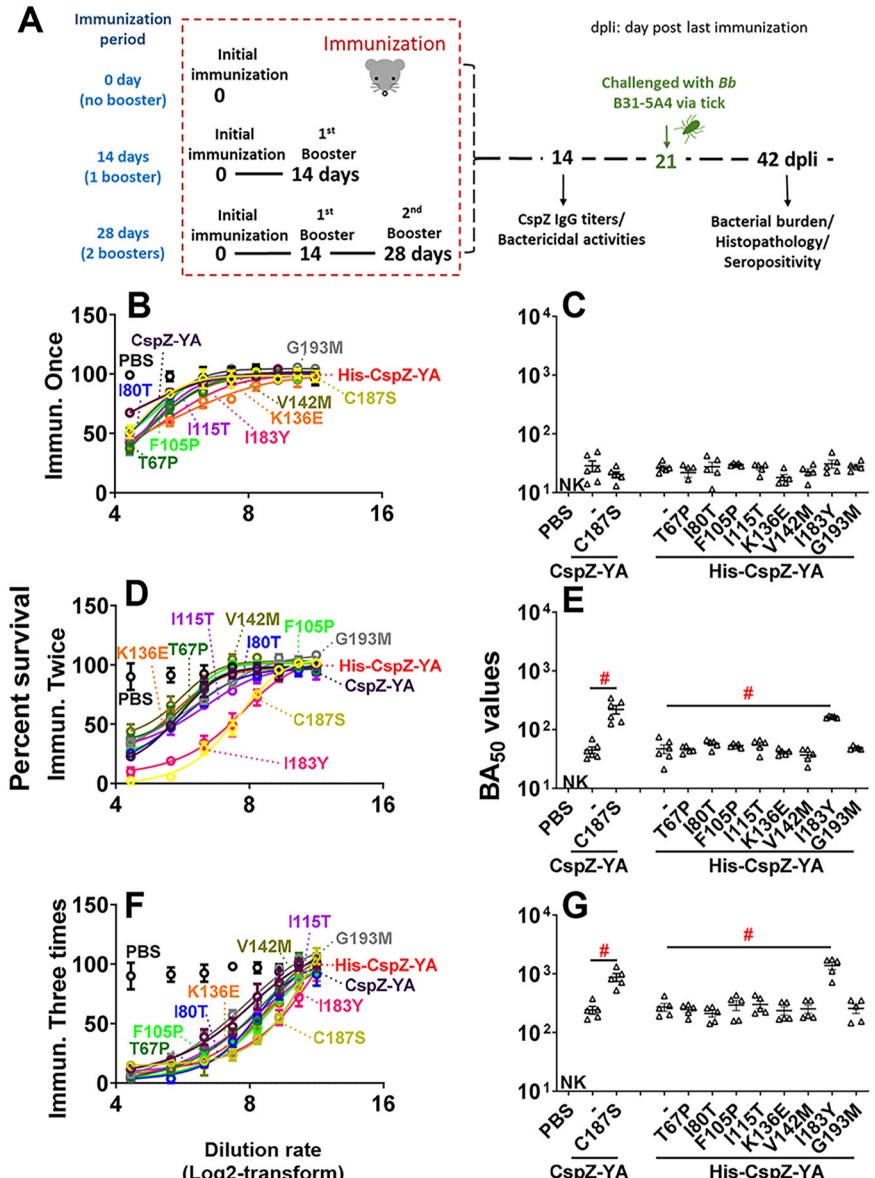

**Fig. 2 | Mice immunized twice and three times with CspZ-YA$_{C187S}$ or CspZ-YA$_{I183Y}$ had sera with more robust levels of borreliacidal activity than CspZ-YA-vaccinated mice. A** The schematic diagram shows pre-adolescent C3H/HeN receiving one inoculation with PBS (control) or the TitierMax Gold (TMG) at the indicated timeframe and frequency, followed by the infection. **B–G** Sera were collected at 14 dpli from pre-adolescent C3H/HeN mice immunized **B, C** once, **D, E** twice, or **F, G** three times. These mice were immunized with PBS (control) or untagged CspZ-YA or histidine-tagged CspZ-YA (His-CspZ-YA), or their mutant proteins. These sera were diluted as indicated, and mixed with guinea pig complement and *B. burgdorferi* B31-A3 for 24 h. Surviving spirochetes were quantified from three fields of view microscopically in three independent experiments. **A, C, E** The survival percentage was derived from the proportion of serum-treated to untreated spirochetes. The data shown are the mean ± SEM of the survival

percentage from three replicates in one representative experiment. **B, D, F** The BA$_{50}$ value, representing the dilution rate that effectively killed 50% of spirochetes, was obtained from curve-fitting and extrapolation of (**A, C, E**). Data shown are the geometric mean ± geometric standard deviation of BA$_{50}$ value from $n = 6$ mice immunized once or twice or $n = 5$ mice immunized three times with CspZ-YA, CspZ-YA$_{C187S}$, or His-CspZ-YA, or $n = 5$ mice inoculated with each of other proteins or PBS and also shown in Table S1. ("NK"), no killing. Statistical significance ($p < 0.05$, Kruskal–Wallis test with the two-stage step-up method of Benjamini, Krieger, and Yekutieli) of differences in borreliacidal titers between groups are indicated ("#"). **E** CspZ-YA: CspZ-YA$_{C187S}$ $p = 0.0005$, His-CspZ-YA: His-CspZ-YA$_{I183Y}$ $p = 0.0032$. **G** CspZ-YA: CspZ-YA$_{C187S}$ $p = 0.0068$, His-CspZ-YA: His-CspZ-YA$_{I183Y}$ $p = 0.0109$. Source data are provided as a Source Data file.

uninfected mice, there was no inflammatory cell infiltration in the joints from CspZ-YA$_{I183Y}$- or CspZ-YA$_{C187S}$-vaccinated mice (Fig. 4). Overall, the mutagenesis of I183Y and C187S allows CspZ-YA vaccination to prevent Lyme disease infection with fewer immunizations.

### The I183Y and C187S mutations did not alter the surface epitopes of CspZ-YA

One hypothesis to address the mechanisms underlying mutagenesis-mediated efficacy enhancement is that CspZ-YA$_{I183Y}$ and CspZ-YA$_{C187S}$

contain novel epitopes, distinct from CspZ-YA. We thus obtained the sera from Lyme disease seropositive patients with elevated levels of CspZ IgGs (36 out of 38 serum samples have elevated levels of CspZ IgGs, Fig. S4). We then tested this hypothesis by comparing the ability of CspZ IgGs in these sera to recognize CspZ-YA, CspZ-YA$_{I183Y}$, and CspZ-YA$_{C187S}$. We found no significant difference between the recognition of CspZ-YA, CspZ-YA$_{I183Y}$, and CspZ-YA$_{C187S}$ (Fig. 5A, two tier pos.), whereas there was minimal detection with sera from seronegative humans in non-endemic areas (Fig. 5A, neg. ctrl.). A

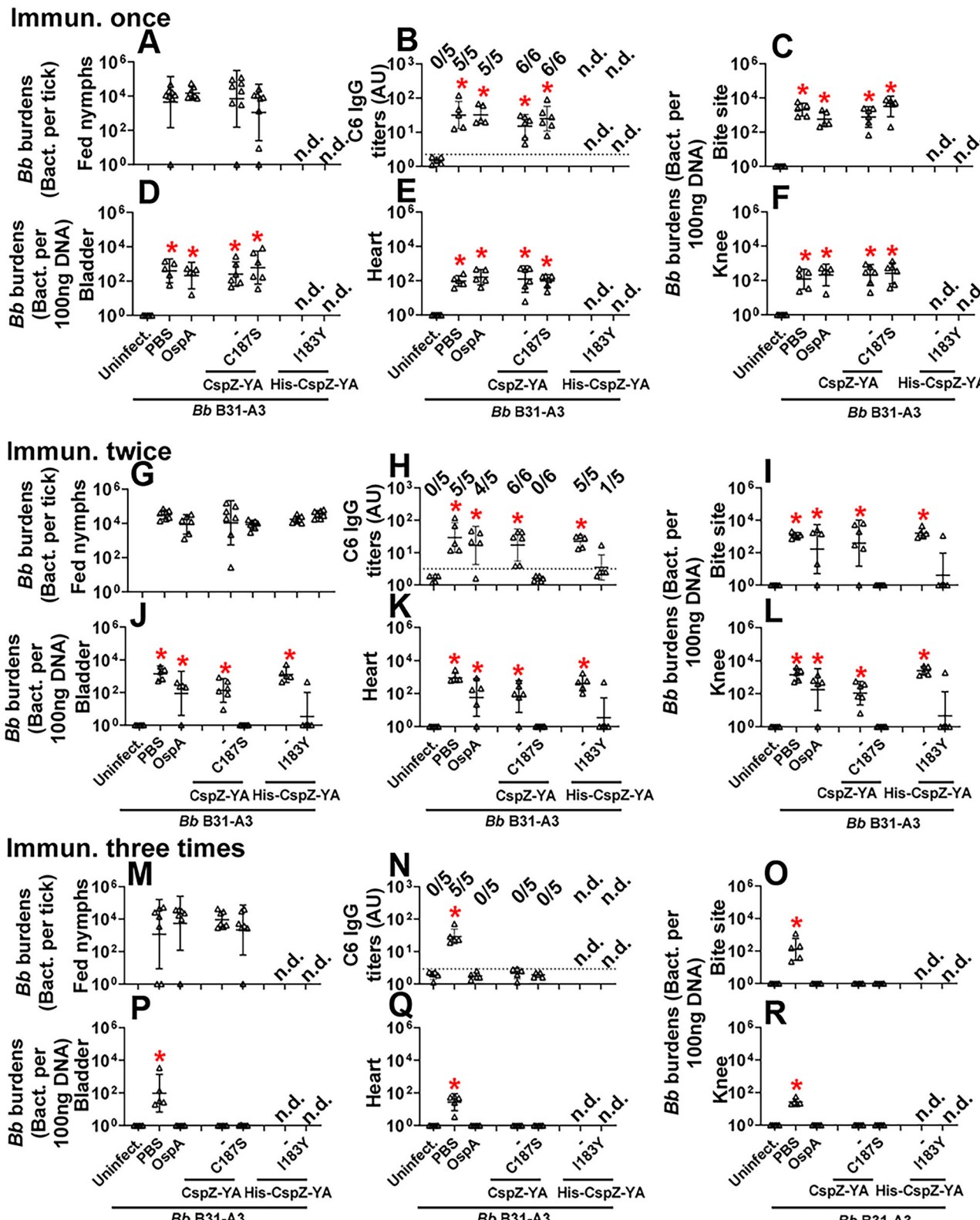

significantly positive correlation was detected for individual patient serum samples to recognize CspZ-YA, CspZ-YA$_{I183Y}$, and CspZ-YA$_{C187S}$ (Fig. 5B–D). We also compared the CspZ IgGs produced in mice immunized twice with CspZ-YA (tagged or untagged), CspZ-YA$_{I183Y}$, and CspZ-YA$_{C187S}$ to recognize each of these proteins in the same fashion. We found similar levels of recognition by CspZ-YA, CspZ-YA$_{I183Y}$, and CspZ-YA$_{C187S}$ for the sera from each immunization group of mice, and such levels of recognition are greater than those from

PBS-inoculated control mice (Fig. 5E). Additionally, when combining the values of recognition from different immunization groups of mice, we observed a significantly positive correlation for those sera to recognize CspZ-YA, CspZ-YA$_{I183Y}$, and CspZ-YA$_{C187S}$ (Fig. 5F–H). Such indistinguishable human or mouse CspZ IgGs recognition by CspZ-YA, CspZ-YA$_{I183Y}$, and CspZ-YA$_{C187S}$ does not support the hypothesis that that mutagenesis of I183Y and C187S changes epitopes of CspZ-YA. We also determined the crystal structure of CspZ-YA$_{C187S}$ at 1.95 Å

**Fig. 3 | Immunizing twice with CspZ-YA$_{C187S}$ or CspZ-YA$_{I183Y}$ but not CspZ-YA protected mice from seroconversion and borrelial tissue colonization.** Pre-adolescent C3H/HeN mice inoculated with PBS or indicated OspA or CspZ-YA proteins or mutant protein (**A**–**F**) once, (**G**–**L**) twice, or (**M**–**R**) three times, followed by infection using nymphs carrying *B. burgdorferi* B31-A3 in the fashion described in Fig. 2A. Mice inoculated with PBS that are not fed on by nymphs were included as an uninfected control group (uninfect.). **B**, **H**, **N** Seropositivity was determined by measuring the levels of IgG against C6 peptides in the sera of those mice at 42 days post-last immunization using ELISA. The mouse was considered seropositive if that mouse had IgG levels against C6 peptides greater than the threshold, the mean plus 1.5-fold standard deviation of the IgG levels against C6 peptides from the PBS-inoculated, uninfected mice (dotted line). The number of mice in each group with anti-C6 IgG levels greater than the threshold (seropositive) is shown. Data shown are the geometric mean ± geometric standard deviation of the titers of anti-C6 IgG from $n = 6$ mice immunized once or twice or $n = 5$ mice immunized with CspZ-YA or CspZ-YA$_{C187S}$, or $n = 5$ mice immunized with each of other proteins or $n = 5$ uninfected mice. Statistical significances ($p < 0.05$, Kruskal–Wallis test with the two-stage step-up method of Benjamini, Krieger, and Yekutieli) of differences in IgG titers relative to (*) uninfected mice are presented. **A**, **C**–**F**, **B**, **H**–**L**, **M**, and **O**–**R** *B. burgdorferi* (*Bb*) burdens at **A**, **G**, **M** nymphs after when feeding to repletion or **C**, **I**, **O** the tick feeding site ("Bite Site"), **D**, **J**, **P** bladder, **E**, **K**, **Q** heart, and **F**, **L**, **R** knees, were quantitatively measured at 42 days post-last immunization,

shown as the number of *Bb* per 100 ng total DNA. Data shown are the geometric mean ± geometric standard deviation of the spirochete burdens from $n = 6$ nymphs feeding on OspA-immunized mice, $n = 7$ nymphs feeding on mice immunized with other proteins, $n = 6$ mice immunized once or twice, or $n = 5$ mice immunized three times with CspZ-YA or CspZ-YA$_{C187S}$, or $n = 5$ mice immunized with each of other proteins, or $n = 5$ uninfected mice. Asterisks indicate the statistical significance ($p < 0.05$, Kruskal–Wallis test with the two-stage step-up method of Benjamini, Krieger, and Yekutieli) of differences in bacterial burdens relative to uninfected mice. **B** Uninfect.: PBS, OspA, CspZ-YA, CspZ-YA$_{C187S}$, $p = 0.0028, 0.0016, 0.0329, 0.0047$. **C** Uninfect.: PBS, OspA, CspZ-YA, CspZ-YA$_{C187S}$, $p = 0.0018, 0.0207, 0.026, 0.0001$. **D** Uninfect.: PBS, OspA, CspZ-YA, CspZ-YA$_{C187S}$, $p = 0.0035, 0.0132, 0.0123, 0.0017$. (**E**) Uninfect.: PBS, OspA, CspZ-YA, CspZ-YA$_{C187S}$, $p = 0.0184, 0.0031, 0.0022, 0.0082$. **F** Uninfect.: PBS, OspA, CspZ-YA, CspZ-YA$_{C187S}$, $p = 0.0457, 0.0074, 0.0035, 0.0008$. **H** Uninfect.: PBS, OspA, CspZ-YA, His-CspZ-YA, $p = 0.002, 0.0063, 0.002, 0.003$. **I** Uninfect.: PBS, OspA, CspZ-YA, His-CspZ-YA, $p = 0.009, 0.0211, 0.0081, 0.0035$. **J** Uninfect.: PBS, OspA, CspZ-YA, His-CspZ-YA, $p = 0.0003, 0.0088, 0.0339, 0.0004$. **K** Uninfect.: PBS, OspA, CspZ-YA, His-CspZ-YA, $p = 0.0002, 0.0491, 0.0475, 0.0035$. **L** Uninfect.: PBS, OspA, CspZ-YA, His-CspZ-YA, $p = 0.0009, 0.0443, 0.0473, 0.0002$. **N** Uninfect.: PBS $p = 0.0071$. **O** Uninfect.: PBS $p = 0.0001$. **P** Uninfect.: PBS $p = 0.0002$. **Q** Uninfect.: PBS $p = 0.0001$. **R** Uninfect.:PBS $p = 0.0001$. Source data are provided as a Source Data file.

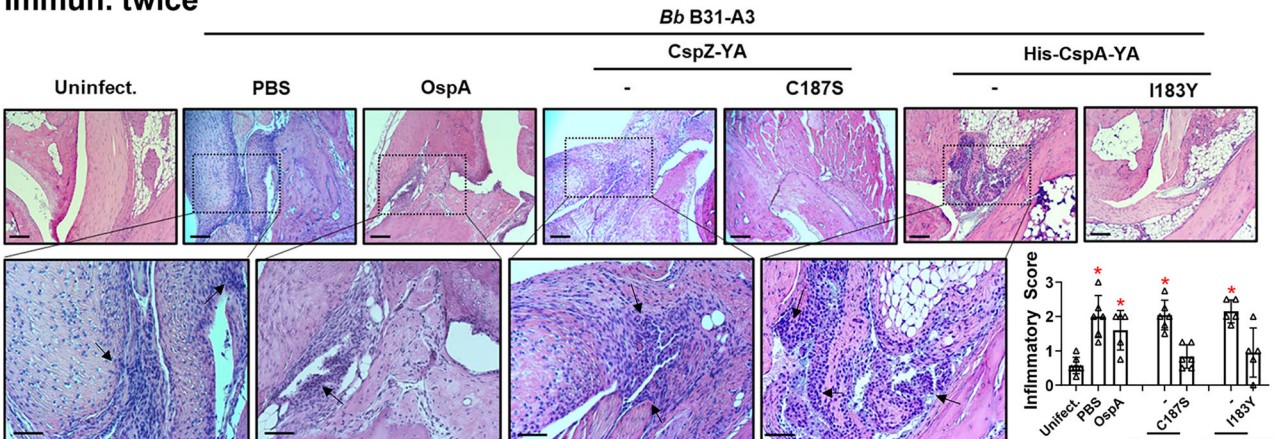

**Fig. 4 | Immunizing twice with CspZ-YA$_{C187S}$ or CspZ-YA$_{I183Y}$ but not CspZ-YA protected mice from Lyme disease-associated joint inflammation.** Pre-adolescent C3H/HeN mice inoculated with PBS or indicated OspA or CspZ-YA proteins or mutant protein were inoculated twice, followed by infection using nymphs carrying *B. burgdorferi* B31-A3 in the fashion described in Fig. 2A. Tibio-tarsus joints at 42 days post-last immunization were collected to assess inflammation by staining these tissues using hematoxylin and eosin. Representative images from one mouse per group are shown. The top panels are lower-resolution images (joint, ×10 [bar, 160 µm]); the bottom panels are higher-resolution images (joint, 2 × 20 [bar, 80 µm]) of selected areas (highlighted in top panels). Arrows indicate infiltration of immune cells. (Inset figure) To quantify inflammation of joint

tissues, at least ten random sections of tibiotarsus joints from each mouse were scored on a scale of 0–3 for the levels of inflammation. Data shown are the mean inflammation score ± standard deviation of the inflammatory scores from $n = 5$ mice immunized with OspA, His-CspZ-YA, or His-CspZ-YA$_{I183Y}$, $n = 6$ mice inoculated with PBS, CspZ-YA, or CspZ$_{C187S}$, or $n = 7$ uninfected mice. Asterisks indicate the statistical significance ($p < 0.05$, Kruskal–Wallis test with the two-stage step-up method of Benjamini, Krieger, and Yekutieli) of differences in inflammation relative to uninfected mice. All images were not cropped. (Inset figure) Uninfect.: PBS, OspA, CspZ-YA, His-CspZ-YA $p = 0.0039, 0.015, 0.0004, 0.0003$. Source data are provided as a Source Data file.

resolution. As the attempt to obtain the crystals for CspZ-YA$_{I183Y}$ was unsuccessful, we used AlphaFold to acquire the predicted structure of this mutant protein[46]. Superimposed structures of CspZ-YA, CspZ-YA$_{C187S}$, and CspZ-YA$_{I183Y}$, using CCP4MG software[39], showed no significant differences in surface epitopes (Fig. 5I). Such superimposed structures included the regions corresponding to known Factor H-binding epitopes, as deduced based on the lack of structural deviations observed in the global fold and side chain orientations (Fig. 5I). This is consistent with the fact that I183 and C187 are not surface-exposed but are buried between helices H and I of CspZ-YA (Fig. 5J). Finally, using putative immunogenic epitope prediction based on the matrix of lowest coupling energies (MLCE)[47], we identified two identical patches of epitopes for CspZ-YA, CspZ-YA$_{C187S}$, and CspZ-

YA$_{I183Y}$: One of these patches involved the N-terminal residues 20 to 26 and the C-terminal residues 232 to 235 while the other patch covered residues 59 to 79 and 119 to 129 (Fig. S5). Notably, since MLCE identifies amino acids associated with flexible hot spots, the terminal regions may have been highlighted as potential epitopes due to their inherent flexibility. Taken together, our structural and immunogenicity results demonstrated no alteration of surface epitopes after the mutation of I183 and C187 in CspZ-YA.

### The I183Y and C187S mutations resulted in enhanced interactions between helices H and I of CspZ-YA proteins
Both I183 and C187 are located on and buried in helix H, raising the possibility that efficacy improvement can be attributed to the

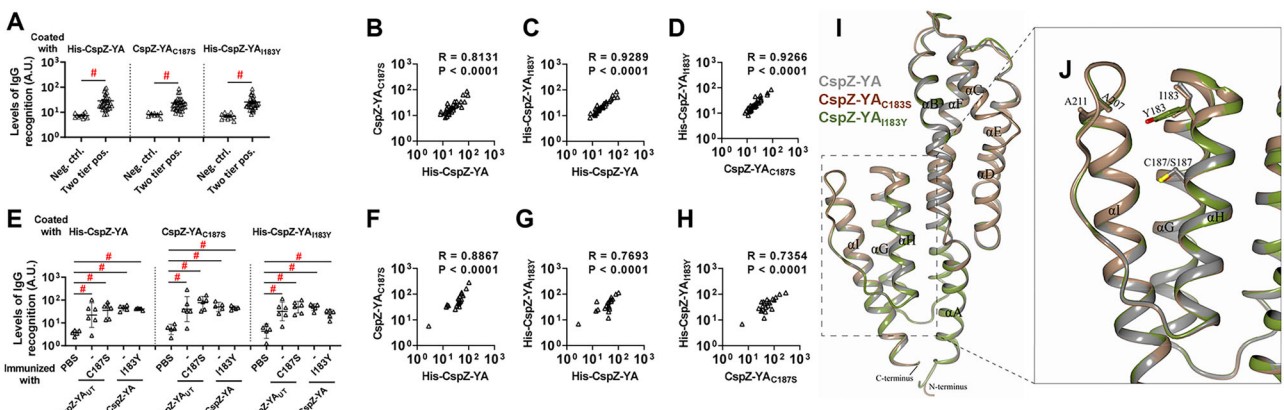

**Fig. 5 | Human and mouse CspZ antibodies recognized CspZ-YA$_{C187S}$ or CspZ-YA$_{I183Y}$ at indistinguishable levels from CspZ-YA. A–D** Patient serum samples ("Two tier positive"; Positive in Lyme disease two tier test) were to determine their levels of recognition to histidine-tagged CspZ-YA (His-CspZ-YA) or CspZ-YA$_{I183Y}$ (His-CspZ-YA$_{I183Y}$), or untagged CspZ-YA$_{C187S}$ or using ELISA. The serum samples from humans residing in non-endemic area of Lyme disease (Neg. ctrl.) were included as control. **A** Data shown are the geometric mean ± geometric SD of levels of recognition from $n = 36$ patient serum samples or $n = 10$ negative control serum samples. Statistical significance ($p < 0.05$, Kruskal–Wallis test with the two-stage step-up method of Benjamini, Krieger, and Yekutieli) of differences in levels of recognition by groups are indicated ("#"). **E–H** Sera from His-CspZ-YA or -CspZ-YA$_{I183Y}$ (I183Y)-, or untagged CspZ-YA- or CspZAC187S (C187S)-immunized C3H/HeN mice (twice immunization, Fig. 2A) were collected at 14dpli. PBS-inoculated mice were included as control. **E** The levels of serum recognition by indicated CspZ-YA proteins were measured by ELISA. Data shown are the geometric mean ± geometric SD of levels of recognition from serum samples of $n = 6$ mice immunized with CspZ-YA or CspZ-YA$_{C187S}$ or $n = 5$ of mice inoculated with other proteins or PBS. Statistical significance ($p < 0.05$, Kruskal–Wallis test with the two-stage step-up

method of Benjamini, Krieger, and Yekutieli) of differences in levels of recognition by groups are indicated ("#"). For **B–D** human or **F–H** mouse sera, the values representing the levels of recognition by **B, F** CspZ-YA vs. CspZ-YAC187S, **C, G** CspZ-YA vs. CspZ-YAI183Y, or **D, H** CspZ-YAC187S vs. CspZ-YAI183Y were plotted. The correlation of these values was determined using Spearman analysis and shown as $R$ values and $P$ values ($p < 0.05$, statistical significance). **I** Superimposed crystal structures of CspZ-YA (gray; PDB ID 9F1V), CspZ-YA$_{C187S}$ (brown; PDB ID 9F21), and the predicted structure of CspZ-YA$_{I183Y}$ (green). All nine α-helices are labeled (αA-αI). **J** Shown is the region in CspZ-YA, CspZ-YA$_{C187S}$, and CspZ-YA$_{I183Y}$ where mutations were introduced. Residues associated with mutations are illustrated as thick bonds. **A** Neg. ctrl.:Two tier pos. for His-CspZ-YA $p < 0.0001$. Neg. ctrl.:Two tier pos. for CspZ-YA$_{C187S}$ $p < 0.0001$. Neg. ctrl.:Two tier pos. for His-CspZ-YA$_{I183Y}$ $p < 0.0001$. **E** PBS: CspZ-YA, CspZ-YA$_{C187S}$, His-CspZ-YA, His-CspZ-YA$_{I183Y}$ for His-CspZ-YA $p = 0.0359, 0.0030, 0.0028, 0.0089$, PBS: CspZ-YA, CspZ-YA$_{C187S}$, His-CspZ-YA, His-CspZ-YA$_{I183Y}$ for CspZ-YA$_{C187S}$ $p = 0.0182$, 0.0002, 0.0031, 0.0079, PBS: CspZ-YA, CspZ-YA$_{C187S}$, His-CspZ-YA, His-CspZ-YA$_{I183Y}$ for His-CspZ-YA$_{I183Y}$ $p = 0.0204, 0.0018, 0.0021, 0.0178$. Source data are provided as a Source Data file.

stabilization and/or enhancement of intramolecular interactions (rather than alteration of the epitopes). We attempted to use protein crystal structures to investigate this possibility by comparing the electron density map surrounding C187 and S187 of CspZ-YA and CspZ-YA$_{C187S}$, respectively. We found a water molecule present in the space between helix I and C187 of CspZ-YA, as well as between helix I and S187 of CspZ-YA$_{C187S}$ (see the arrow in Fig. 6A, B). That water molecule does not impact the intramolecular interactions in CspZ-YA (Fig. 6A), but coordinates the interactions with S187 on the helix H and E214 on the helix I via hydrogen bonding in CspZ-YA$_{C187S}$(dotted lines in Fig. 6B), likely enhancing the protein stability. Additionally, the three-dimensional structure of CspZ revealed a hydrophobic core between helices G, H and I where residues Y207 and Y211 are on one side, along with F176, I183 and F217 on the other side (Fig. 6C). Unlike CspZ, a cavity was found in CspZ-YA due to the replacement of two bulky and non-polar amino acid residues, Y207 and Y211, with alanine (the red highlight in Fig. 6D). Further, the altered conformation in CspZ-YA prevents R206 from interacting with E186, exacerbating the cavity-mediated structural instability (Fig. 6D). Unlike CspZ-YA, we found that the AlphaFold predicted structure of CspZ-YA$_{I183Y}$ showed that the cavity is filled by a bulky, non-polar residue (i.e., Y183), restoring the hydrophobic core between helix H and I (Fig. 6E). This is consistent with the intention of achieving a more stabilized structure of CspZ-YA$_{I183Y}$ (Fig. 1B).

We next compared the structural flexibility of CspZ, CspZ-YA, CspZ-YA$_{C187S}$ and CspZ-YA$_{I183Y}$ using MD simulations, which allowed us to obtain the root mean square deviation (RMSD) value, the distance between the Cα atoms from a reference structure during the 300 nanoseconds (ns) of the simulation at 300 K (26 °C) (Videos S1–S4). We found that all these proteins showed similar RMSD values (over the three simulations ranging from 0.8 to 3.3 Å in wild-type CspZ, 1.2 to

3.8 Å in CspZ-YA, 1.0 to 3.5 Å in CspZ-YA$_{C187S}$, and 0.9 to 3.7 Å in CspZ-YA$_{I183Y}$ (Fig. 7A)). These results indicate similar flexibility of the overall conformation for CspZ, CspZ-YA, CspZ-YA$_{C187S}$ and CspZ-YA$_{I183Y}$. We further applied these proteins to the root mean square fluctuation (RMSF) analyses to assess the flexibility of individual residues during the simulation and observed five regions with increasing dynamics (greater flexibility) in CspZ, CspZ-YA, CspZ-YA$_{C187S}$, and CspZ-YA$_{I183Y}$. These regions include R1 (the loop between helices B and C; residues 63–70), R2 (the loop between helices E and F; residues 120–128), R3 (the loop between helices F and G; residues 157–160), R4 (the loop between helices G and H (residues 180–182), and R5 (the loop between helices H and I; residues 198-211) (Fig. 7B, C). The RMSF values of R1, R2, R3, or R4 among CspZ, CspZ-YA, CspZ-YA$_{C187S}$, and CspZ-YA$_{I183Y}$ are similar (Fig. 7B). However, the RMSF values of R5 varied among these tested proteins with CspZ-YA having the highest flexibility (0.9–5.5 Å), followed by CspZ-YA$_{C187S}$ (0.9–5.0 Å), CspZ-YA$_{I183Y}$ (0.9–4.7 Å), and CspZ (0.9–2.2 Å) (Fig. 7B). Such MD simulations were also performed at 310 K (37 °C), showing similar results (data now shown). Thus, the results here identified a less flexible loop region between helices H and I (R5; residues 198–221) in the CspZ-YA$_{C187S}$ and CspZ-YA$_{I183Y}$, compared to CspZ-YA. Overall, these findings support the notion that I183Y and C187S mutagenesis facilitated the helix H–I interactions in CspZ-YA proteins.

**The I183Y and C187S mutations promoted the stability of the CspZ-YA epitopes recognized by CspZ-targeting, Lyme borrelia-killing monoclonal antibodies**

The enhanced intramolecular interactions by I183Y and C187S mutagenesis raise the possibility of CspZ-YA$_{I183Y}$ and CspZ-YA$_{C187S}$ having increased stability. We thus examined whether CspZ-YA$_{I183Y}$ and CspZ-YA$_{C187S}$ have greater thermostability than CspZ-YA by incubating them

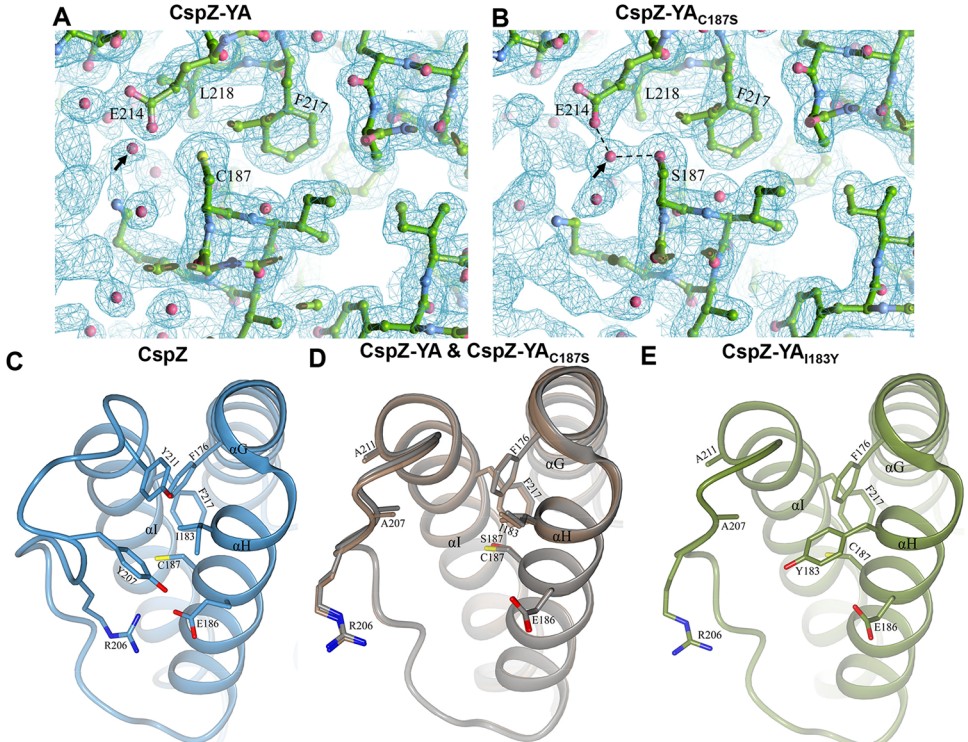

**Fig. 6 | The comparison of CspZ, CspZ-YA, CspZ-YA$_{C187S}$, and CspZ-YA$_{I183Y}$ structures suggests the helix H–I interactions impacted by the C187S and I183Y mutagenesis.** The structures here were obtained from CspZ (PDB ID 9F7I), CspZ-YA (PDB ID 9F1V), CspZ-YA$_{C187S}$ (PDB ID 9F21), and AlphaFold predicted structure of CspZ-YA$_{I183Y}$. A portion of the electron density map from the newly solved structure of CspZ-YA$_{C187S}$ is shown in Fig. S8B. **A, B** Shown is the 2Fo-Fc electron density map contoured at 1σ of the region around C187 in (**A**) CspZ-YA and S187 in (**B**) CspZ-

YA$_{C187S}$. The hydrogen bond formed between the water molecule with S187 and E214 were highlighted. (**C–E**) The crystal structures of **C** CspZ from *B. burgdorferi* B31, **D** superimposed CspZ-YA (gray) and CspZ-YA$_{C187S}$ (brown) structures, and **E** the predicted structure of CspZ-YA$_{I183Y}$ show the hydrophobic core accounting for helices G, H and I and the residues Y207, Y211, I183, and C187 in CspZ and the equivalent residues in CspZ-YA and CspZ-YA$_{I183Y}$.

at elevated temperatures and determining the temperature that 50% of these proteins are denatured (the melting temperature (Tm)). We found indistinguishable Tm-values between CspZ-YA$_{I183Y}$ and CspZ-YA$_{C187S}$ (61.8 and 62.7 °C, respectively) (Fig. 8A and Table S3). In contrast, the Tm-values of CspZ-YA were significantly lower, 57.5 and 58.4 °C for histidine-tagged and untagged CspZ-YA, respectively, indicating a stability enhancement through mutagenesis (Fig. 8A and Table S3). We next examined the possibility of the I183Y and C187S mutations to enhance the long-term stability of the protective epitopes in the CspZ-YA structures. To test this possibility, we generated humanized, recombinant, chimeric, and monoclonal CspZ-YA IgGs that contain the Fc region of human IgG1 and F(ab')2 from 1139 or 1193, our two monoclonal CspZ-YA mAbs documented to eliminate Lyme borreliae[38]. The humanized IgGs, 1139c and 1193c, bound purified CspZ-YA in a surface plasmon resonance (SPR) experiment with K$_D$-values of 24.2 and 157 nM, respectively (Fig. S6A, B). These antibodies were also shown to block the FH-binding ability of CspZ (Fig. S6C) and promote bacterial lysis (Fig. S6D) and opsonophagocytosis of *B. burgdorferi* (Fig. S6E). We incubated CspZ-YA$_{I183Y}$, CspZ-YA$_{C187S}$, or CspZ-YA (untagged and histidine-tagged) at 4 or 37 °C for different periods of time and then examined the ability of 1139c or 1193c to bind to each of these CspZ-YA proteins. We found all CspZ-YA proteins or variants previously incubated at 4 °C for 6- or 24-h or at 37 °C for 6-h displayed similar levels of recognition to these proteins prior to incubation (Fig. 8B, C). However, CspZ-YA but not CspZ-YA$_{I183Y}$ and CspZ-YA$_{C187S}$ previously incubated at 37 °C for 24-h had 1.6-fold lower levels of recognition, compared to those proteins prior to incubation (Fig. 8B, C), even though SEC-HPLC did not indicate significant aggregation or degradation (results not shown). These findings demonstrated long-term stability enhancement of CspZ-YA proteins in

the physiological temperature by I183Y and C187S mutagenesis, specifically on the structures that promote protective antibody induction.

## Discussion

Although many native microbial antigens are effective as immunogens, some of them present challenges for vaccine development, including a lack of immunogenicity or protective efficacy[4] or potential toxicities, leading to safety concerns[48–50]. Several strategies of antigen engineering have been proposed and tested to optimize the immunogen, (for review, see refs. 51–53). One such approach is by unmasking normally occluded epitopes to enhance protective antibody responses. Taking CspZ as an example, immunizing the native version of such a protein did not prevent Lyme disease infection[31,37]. However, we previously generated CspZ-YA through the mutagenesis of Y207A and Y211A and found that CspZ-YA can induce robust levels of borreliacidal antibodies that prevent the infection, suggesting a newly generated surface epitopes in CspZ-YA[36,37]. In this study, the comparison of our newly obtained three-dimensional structure of CspZ-YA with the previously resolved structure of the CspZ-FH complex (PDB 9F71), explains the inability of CspZ-YA to bind to FH and identifies exposed epitopes surrounding the FH-binding sites[40]. Thus, the protective antibodies could be triggered by vaccination with CspZ-YA. We envision that such antibodies would block the native antigen, CspZ, on *B. burgdorferi* from binding to factor H, immediately after *B. burgdorferi* invades the hosts. Alternatively, protective antibodies with higher affinity (lower $K_D$ values; $10^{-7}$ to $10^{-8}$ M in Fig. S6) than factor H for binding to CspZ (~mid-$10^{-7}$ M[40]) may replace the FH in FH-bound CspZ. Overall, using CspZ as a model, our work here demonstrated the potential mechanisms of unmasking protective epitopes to promote immunogenicity.

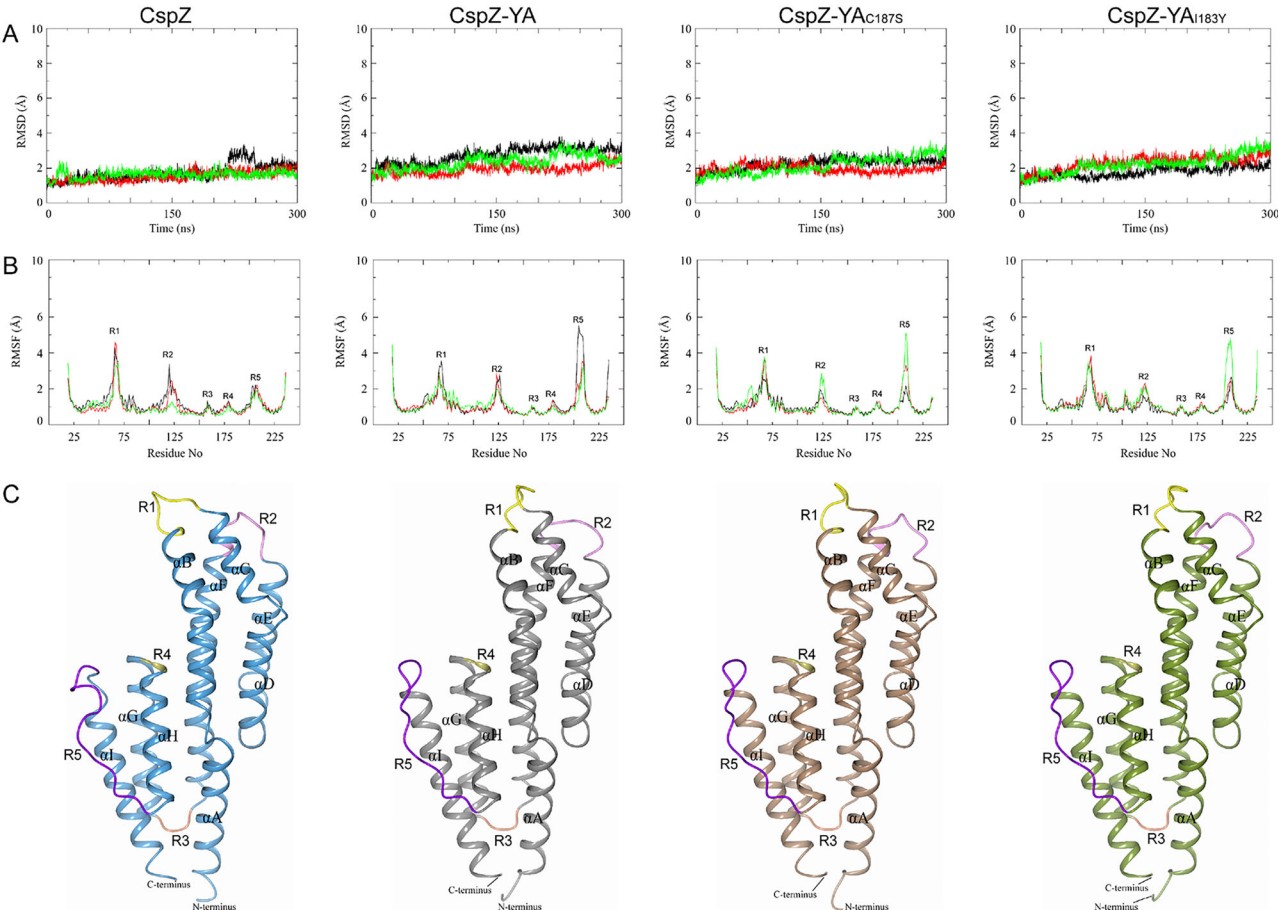

**Fig. 7 | MD simulations of CspZ, CspZ-YA, CspZ-YA$_{C187S}$, and CspZ-YA$_{I183Y}$ show flexibility of the loop between helices H and I impacted by C187S and I183Y mutagenesis. A** The structural drift of CspZ, CspZ-YA, CspZ-YA$_{C187S}$, and CspZ-YA$_{I183Y}$ is shown over 300 ns at 300 K. **B** The RMSF measurements of Cα atoms for CspZ, CspZ-YA, CspZ-YA$_{C187S}$, and CspZ-YA$_{I183Y}$ over 300 ns at 300 K are plotted against the residue number. Different colors (black, green, red) represent three simulation runs. **C** The crystal structures of wild-type CspZ (PDB ID 9F7I), CspZ-YA (PDB ID 9F1V), CspZ-YA$_{C187S}$ (PDB ID 9F21), and the predicted structure of CspZ-YA$_{I183Y}$ illustrate the five regions (R1-R5) with increased RMSF values. All nine α-helices of CspZ variants are labeled (αA-αI).

The other commonly used antigen engineering approach for vaccine development is to mutagenize the amino acid residues based on the three-dimensional antigen structures with the goal of enhancing antigen stability by promoting suitable intramolecular interactions[6,7]. This approach can potentially lead to improved immunogenicity and efficacy[52,53]. One of these mutagenesis strategies is "cavity filling" to fill the hydrophobic cores for stability increase (e.g., stabilizing the perfusion structures of a respiratory syncytial virus (RSV) F protein or the binding interface of HIV glycoproteins (gp120-gp41)[54,55]. In Lyme borreliae CspZ proteins, the mutagenesis of Y207A and Y211A reduced the hydrophobic interactions and created a cavity at the C-terminal H/I loop and helix H of CspZ-YA (Fig. 6D), destabilizing the protein's conformation. Introduction of hydrophobic amino acid residues (i.e., isoleucine, leucine, valine, phenylalanine, and tyrosine) ideally would fill the cavity to maintain protein stability[56]. In this study, we included the I183Y mutation for cavity filling (Fig. 6E). This strategy proves the cavity-filling concept in a Lyme borreliae antigen by elevating the efficacy of CspZ-YA-triggered bactericidal antibodies and preventing bacterial colonization and disease manifestations at lower immunization frequency.

Another structural modification that can be made to improve antigen properties is the manipulation of disulfide bonds. Adding cysteine residues may lead to the formation of disulfide bonds, promoting intramolecular interaction and protein stability[57,58]. However, the free cysteine residue in the antigens may also contribute to unwanted intermolecular interactions, resulting in protein aggregations[41,42]. Here, we attempted to test the impact of C187 on CspZ-YA in protein stability by replacing this residue with serine. Our structural evidence further attributes such stability enhancement to the role of water molecule-coordinated helix H−I intramolecular interactions. In fact, sulfur had a slightly greater van der Waals radius (-1.8 Å), compared to oxygen (-1.5 Å). Thus, the highly polar nature of serine together with a slightly smaller footprint of oxygen from S187 may be the cause of the water molecule positioning in the fashion to enhance hydrogen bond-mediated intramolecular interactions in CspZ-YA$_{C187S}$ (Fig. 6A, B). Overall, this work provides structural evidence suggesting cysteine mutagenesis-mediated stability enhancement. Further, I183 and C187 are located on helix H and are involved in helix H−I interactions. Although they are not located in the FH-binding interface of CspZ, the N-terminus of helix H and the loop H/I are within and immediately adjacent to the FH-binding sites[40,59]. Therefore, it is conceivable that the strengthening of helix H−I interactions in CspZ-YA vaccines stabilizes the structures of the protective epitopes within or adjacent to the FH-binding site, triggering greater levels of protective antibodies. Testing this possibility would require the elucidation of the three-dimensional complexed structure of CspZ-YA and those CspZ-targeted and protective monoclonal antibodies (i.e., 1139c and 1193c).

We found that our CspZ-targeted and protective monoclonal antibodies recognize CspZ-YA$_{I183Y}$ and CspZ-YA$_{C187S}$, better than

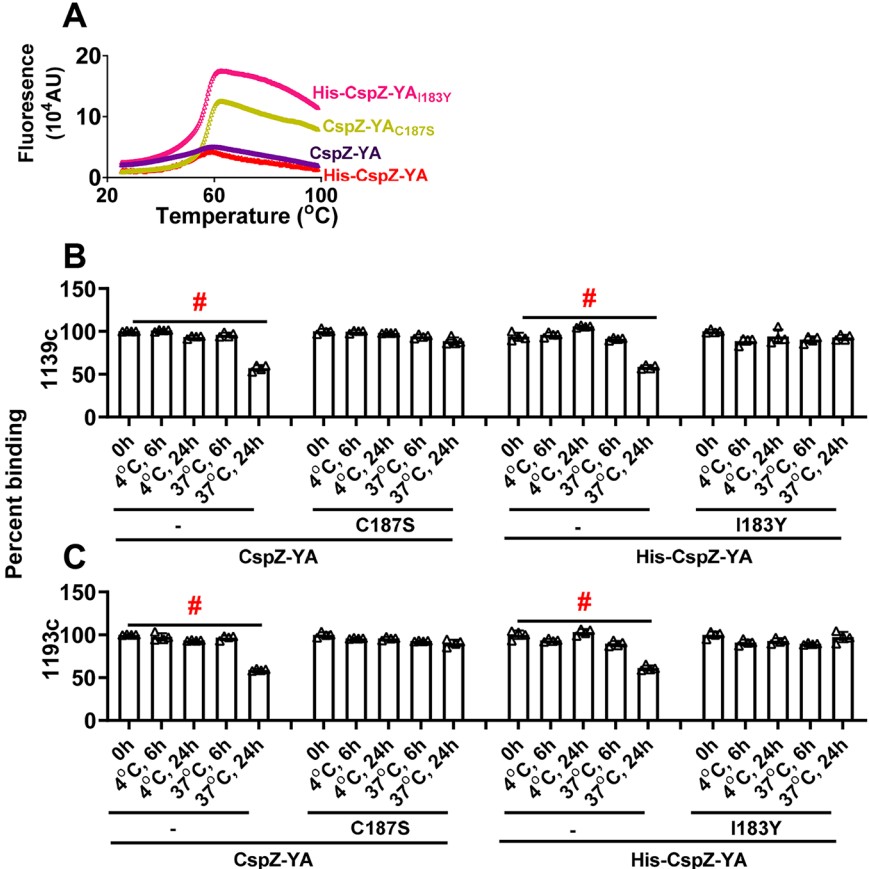

**Fig. 8 | CspZ-YA$_{C187S}$ and CspZ-YA$_{I183Y}$ maintained the recognition by protective CspZ IgGs at higher temperatures for a longer time. A** Untagged or histidine-tagged CspZ-YA or mutant proteins (10 μM) in PBS buffer were subjected to the thermal shift assays with the emission wavelength of 623 nm, as described in the materials and methods section. Shown are the fluorescence intensities of each of the CspZ-YA proteins under temperatures ranging from 25 to 99 °C from one representative experiment. The melting temperature (Tm) was extrapolated from the maximal positive derivative values of the fluorescence intensity (d(RFU)/dT) and shown in Table S3. **B**, **C** One μg indicated CspZ-YA proteins was incubated at 4 or 37 °C for 6- or 24-h prior to being coated on microtiter plate wells. The microtiter plate wells immobilized with each of these proteins before incubation (0-h) were included as unincubated control. The ability of the CspZ monoclonal IgG, **B** 1139c or **C** 1193c, to recognize each of these CspZ-YA proteins were determined using ELISA

in the section "Accelerated stability study" in Materials and Methods. The work was performed on four independent experiments (one replicate per experiment). Data are expressed as the percent binding, derived by normalizing the levels of bound 1139c or 1193c from the wells coated with each of the CspZ-YA proteins in different incubating conditions to that in the unincubated control. Data shown are the mean ± standard deviation of the percent binding of 1139c or 1193c from $n = 4$ experiments. Statistical significance ($p < 0.05$, Kruskal–Wallis test with the two-stage step-up method of Benjamini, Krieger, and Yekutieli) of differences in percent binding between groups are indicated ("#"). **B** CspZ-YA 0 h: CspZ-YA 37 °C, 24 h $p = 0.0019$. His-CspZ-YA 0 h: His-CspZ-YA 37 °C, 24 h $p = 0.0484$. **C** CspZ-YA 0 h: CspZ-YA 37 °C, 24 h $p = 0.0003$. His-CspZ-YA 0 h: His-CspZ-YA 37 °C, 24 h $p = 0.0028$. Source data are provided as a Source Data file.

CspZ-YA at a higher temperature and for longer (i.e., 37 °C for 24 h). As the normal core temperature of mammals is close to 37 °C, both mutations would allow CspZ-YA to persist in a conformation that would promote the continuous production of protective antibodies, thus increasing its suitability for vaccine development. Additionally, the more stable structures suggested by structure-based vaccine design would potentially aid vaccine production and ease required conditions for transportation and storage, promoting the commercialization plan[60]. In this study, we mutagenized CspZ-YA as a model to test the concept of structure-based vaccine design to decrease the minimal immunization frequency that still protects. The results elucidate the mechanisms underlying such a concept using a Lyme disease subunit vaccine as a model. The fact that, in this study, CspZ-YA$_{I183Y}$ or CspZ-YA$_{C187S}$ each allow lower protective immunization frequency makes them attractive for future vaccine development. Finally, the advancement of vaccine design as a result of the recent pandemic underscores the importance of structure-guided approaches for the efficacious optimization of vaccines. This proof-of-concept study thus provides mechanistic insights into

structural vaccinology and illustrates the possibility of revisiting the previously tested but non-efficacious antigens for future vaccine development.

## Methods

### Ethics statement

All mouse experiments were performed in strict accordance with the provisions of the Animal Welfare Act, the Guide for the Care and Use of Laboratory Animals, and the PHS Policy on Humane Care and Use of Laboratory Animals. The protocol (Docket Number 22-451) was approved by the Institutional Animal Care and Use Agency of the Wadsworth Center, New York State Department of Health. All efforts were made to minimize animal suffering. This study also involves secondary use of deidentified archival patient sera collected in previous studies and was approved by the Institutional Review Board (IRB) of New York State Department of Health and Baylor College of Medicine under protocol 565944-1 and H-46178, respectively. Analysis of deidentified patient data was carried out under a waiver of consent.

## Mouse, ticks, bacterial strains, hybridoma, and human serum samples

Four-week-old, female C3H/HeN mice were purchased from Charles River (Wilmington, MA, USA). Although such an age of the mice has not reached sexual maturity, the underdevelopment of the immune system in this age of mice would allow such mice to be more susceptible to Lyme borreliae infection, increasing the signal-to-noise ratio of the readout. That will also provide more stringent criteria to define the protectivity. The *Ixodes scapularis* tick larvae were obtained from BEI Resources (Manassas, VA), and the BALB/c C3-deficient mice to generate *B. burgdorferi*-infected nymphal ticks were from in-house breeding colonies[61]. All mice were housed in the facility with temperature as 65–75 °F (-18–23 °C) and 40–60% humidity, as well as 12-h dark/12-h light cycle. *Escherichia coli* strain BL21(DE3) (EMD Millipore, Burlington, MA) harboring recombinant expression plasmids were grown at 37 °C or other appropriate temperatures in Luria-Bertani broth or agar, supplemented with kanamycin (50 µg/mL). *B. burgdorferi* strain B31-A3 was grown at 33 °C in BSK II complete medium[62]. Cultures of *B. burgdorferi* B31-A3 were tested with PCR to ensure a full plasmid profile before use[63]. Hybridoma that produce the monoclonal antibodies #1139c or #1193c were cultivated in RPMI 1640 medium (Thermo Fisher Scientific, Waltham, MA) containing 10% FBS (Thermo Fisher Scientific) at 37 °C with 5% $CO_2$. Thirty-eight deidentified two-tiered positive human serum samples were obtained from the New York State Department of Health. These serum samples were previously collected from humans who had been tested positive in two-tiered assays, which is the serological definition of Lyme disease infection[64]. The negative control human sera were collected from ten individuals residing in a non-endemic area for Lyme disease.

## Cloning, expression, and purification of OspA, CspZ, CspZ-YA, and CspZ-YA-derived mutant proteins

The DNA encoding histidine-tagged CspZ, CspZ-YA, and CspZ-YA-derived mutant proteins (Table S4) were codon-optimized based on *E. coli* codon usage preference and synthesized by Synbiotech (Monmouth Junction, NJ), followed by subcloning into the pET28a using BamHI/SalI restriction sites. These plasmids were transformed into *E. coli* BL21 (DE3). The DNA encoding untagged CspZ and its derived mutant proteins were codon-optimized based on *E. coli* codon usage preference, synthesized, and subcloned into the pET41a using *Nde*I/*Xho*I sites by GenScript (Piscataway, NJ). These plasmids were transformed into *E. coli* BL21 (DE3). The recombinant protein expression in Luria-Bertani broth at 37 °C was induced with 1 mM Isopropyl-β-D-1-thiogalactopyranoside (IPTG) (EMD Millipore). Once expression was confirmed, the clone with the highest expression for each construct was selected to create glycerol seed stocks. The histidine-tagged CspZ and its mutant derivatives were generated using affinity chromatography as described previously[38]. Untagged CspZ-YA and CspZ-YA$_{C187S}$ were purified by anion exchange chromatography, followed by size exclusion chromatography, as described[65]. Because lipidation is required for recombinant OspA proteins to protect mice from Lyme disease infection[66], the lipidated OspA was included as a control. To generate the lipidated OspA, the previous process was followed[67]. For structural studies, the encoding regions of CspZ-YA and CspZ-YA$_{C187S}$ were cloned into the pETm-11 expression vector (EMD Millipore) containing an N-terminal 6xHis tag, followed by a tobacco etch virus (TEV) protease cleavage site. Both proteins were expressed in *E. coli* BL21 (DE3) and purified by affinity chromatography as described previously for CspZ[68].

## Generation of humanized chimeric CspZ-YA antibodies, #1139c and #1193c

Protective mouse monoclonal antibodies (mAbs) 1139 and 1193 against CspZ were developed previously[38]. These two mAbs were further humanized and chimerized using the service provided by GenScript

Probio (Piscataway, NJ). Briefly, DNA sequencing was performed using the hybridoma to identify the gene coding the variable domain of mAbs 1139 and 1193. Such genes were then grafted with the one coding for human IgG1. The two chimeric mAbs (1139c and 1193c) were then transiently produced in CHO-K1 cells (ATCC, Manassas VA), followed by purification with Protein A affinity chromatography (Cytiva, Marlborough, MA).

## Circular dichroism (CD) spectroscopy

CD analysis was performed on a Jasco 810 spectropolarimeter (Jasco Analytical Instrument, Easton, MD) under nitrogen. CD spectra were measured at room temperature (RT, 25 °C) in a 1 mm path-length quartz cell. Spectra of each of the CspZ-YA proteins (10 µM) were recorded in phosphate-based saline buffer (PBS) at RT, and three far-UV CD spectra were recorded from 190–250 nm in 1 nm increments for far-UV CD. The background spectrum of PBS without proteins was subtracted from the protein spectra. CD spectra were initially analyzed by the software Spectra Manager Program (Jasco). Analysis of spectra to extrapolate secondary structures was performed using the K2D3 analysis programs[69].

## Mouse immunization and infection

C3H/HeN mice were immunized as described, with slight modifications[37]. Fifty µl of PBS (control) or 25 µg of untagged or histidine-tagged CspZ-YA or its mutant proteins, or untagged, lipidated OspA in 50 µl of PBS was thoroughly mixed with 50 µl TiterMax Gold adjuvant (Norcross, GA, USA), resulting in total 100 µl of the inoculum. This inoculum was introduced into C3H/HeN mice subcutaneously once at 0 days post initial immunization (dpii), twice at 0 and 14 dpii, or three times at 0, 14, and 28 dpii (Fig. 2A). At 14 days post-last immunization (dpli), blood was collected via submandibular bleeding to isolate serum for the determination of ability in recognizing CspZ, CspZ-YA and the mutant proteins derived from CspZ-YA, as described in the sections of "ELISAs" and "Borreliacidal assays", respectively (Fig. 2A). At 21 dpli, *B. burgdorferi* B31-A3-infected flat nymphs were placed in a chamber on the immunized or PBS-inoculated C3H/HeN mice as described (Fig. 2A)[70]. Five nymphs were allowed to feed to repletion on each mouse, and a subset of nymphs was collected pre- and post-feeding. At 42 dpli, tick bite sites of skin, bladder, knees, and heart were collected to determine the bacterial burdens, and ankles were also collected at 42 dpli to determine the levels of inflammation described in the section "Quantification of spirochete burdens and histological analysis of joint inflammation (Fig. 2A)." At this time point, blood was also collected via cardiac puncture bleeding to isolate serum for the determination of seropositivity described in the section "ELISAs" (Fig. 2A).

For the mice inoculated with humanized chimeric monoclonal IgGs, C3H/HeN mice were inoculated as described, with slight modifications[38]. C3H mice were intraperitoneally inoculated with irrelevant human IgG (control; Sigma-Aldrich, St. Louis, MO, catalog number AG100), #1139c or #1193c (1 mg/kg) (Fig. S7A). Five mice per group were used in this study. At 24 h after inoculation, five nymphs carrying *B. burgdorferi* strain B31-A3 were allowed to feed to repletion on each mouse, and a subset of nymphs was collected pre- and post-feeding as described[37,61]. Mice were sacrificed at 21 days post-feeding (dpf) to collect the biting site of skin, bladder, knees, and heart to determine the bacterial burdens described in the section "Quantification of spirochete burdens and histological analysis of joint inflammation (Fig. S7A)." Blood was also collected via cardiac puncture bleeding to isolate sera for the determination of seropositivity described in the section "ELISAs" (Fig. S7A).

## ELISAs

To measure the titers of anti-CspZ IgG in the serum samples (Fig. S3), one µg of histidine-tagged CspZ was coated on ELISA plate wells as

described[37]. To determine the ability of anti-CspZ IgG in the sera to recognize CspZ-YA, CspZ-YA$_{I183Y}$, and CspZ-YA$_{C187S}$ (Fig. 5A), each of these proteins with histidine tags (1 μg) was coated on ELISA plate wells as in the same fashion. The procedures following the protein coating are as described previously[37]. For each serum sample, the maximum slope of optical density/minute of all the dilutions of the serum samples was multiplied by the respective dilution factor, and the greatest value was used as an arbitrary unit (AU) to represent the antibody titers for the experiment to obtain anti-CspZ IgG (Fig. S3) or the ability of the anti-CspZ IgG in the sera to recognize CspZ-YA and different CspZ-YA mutant proteins (Fig. 5A). The quality of the correlation for the ability of those CspZ IgG in recognizing CspZ-YA vs. CspZ-YAC187S, CspZ-YA vs. CspZ-YAI183Y, or CspZ-YAC187S vs. CspZ-YAI183Y was determined by the R and P values of Spearman analysis, which was calculated using dose-response stimulation fitting in GraphPad Prism 9.3.1.

Additionally, the seropositivity of the mice after infection with *B. burgdorferi* was determined by detecting the presence or absence of the IgGs that recognize C6 peptides (Fig. 3B, H. N). This methodology has been commonly used for human Lyme disease diagnosis[71] and performed as described in our previous work[38]. For each serum sample, the maximum slope of optical density/minute of all the dilutions was multiplied by the respective dilution factor, and the greatest value was used as representative of anti-C6 IgG titers (arbitrary unit (AU)). The seropositive mice were defined as the mice with the serum samples yielding a value greater than the threshold, the mean plus 1.5-fold standard deviation of the IgG values derived from the uninfected mice.

We also determined the ability of #1139c or #1193c to prevent FH from binding to CspZ (Fig. S6C), which was performed as described previously with modifications[38]. Each ELISA microtiter well was coated with one μg of histidine-tagged CspZ. After being blocked with 5% BSA in PBS buffer, the wells were incubated with PBS (control) or serially diluted irrelevant human IgG (Human IgG isotype control) #1139c or #1193c (0.4, 0.8, 1.6, 3.125, 6.25, 12.5, 25, and 50 nM) followed by being mixed with 500 nM of human FH. Sheep anti-human FH (1:200×, Thermo Fisher Scientific, catalog number: SHAHU-FH) and then donkey anti-sheep HRP (1:2000×, Thermo Fisher Scientific, catalog number: A16041) were added, and the levels of FH binding were detected by ELISA as described previously[38]. Data were expressed as the proportion of FH binding from serum-treated to PBS-treated wells. The 50% inhibitory concentration (IC$_{50}$) (the inlet figure of Fig. S6C), representing the IgG concentration that blocks 50% of FH binding, was calculated using dose-response stimulation fitting in GraphPad Prism 9.3.1.

## Borreliacidal assays

The ability of serum samples (Fig. 2B–G) or humanized chimeric monoclonal CspZ IgG (#1139c and #1193c, Fig. S6D) to eradicate *B. burgdorferi* B31-A3 was determined as described with modifications[36,37]. Briefly, the sera collected from mice immunized with different CspZ-YA proteins at different immunization frequencies were heat-treated to inactivate complement. Each of these serum samples, #1139c or #1193c, was serially diluted, and mixed with complement-preserved or heat-inactivated guinea pig serum (Sigma-Aldrich, negative control). After adding *B. burgdorferi* B31-A3, the mixture was incubated at 33 °C for 24 h. Surviving spirochetes were quantified by directly counting the motile spirochetes using dark-field microscopy and expressed as the proportion of serum-treated to untreated Lyme borreliae. The 50% borreliacidal activities (BA$_{50}$), representing the serum dilution rate (for the serum samples in Fig. 2B–G) or the concentration of IgGs (for #1139c and #1193c in Fig. 2B–G) that kills 50% of spirochetes, was calculated using dose-response stimulation fitting in GraphPad Prism 9.3.1.

## Quantification of spirochete burdens and histological analysis of joint inflammation

DNA was extracted from the indicated mouse tissues to determine the bacterial burdens (Fig. 3C–F, I–L, and O–R), using quantitative PCR analysis as described[37]. Note that spirochete burdens were quantified based on the amplification of *recA* using the forward and reverse primers with the sequences: recAfp: 5′-GTGGATCTATT GTATTAGATGAGGCTCTCG-3′ and recArp: 5′-GCCAAAGTTCT GCAACATTAACACCTAAAG-3′, respectively. The number of *recA* copies was calculated by establishing a threshold cycle (Cq) standard curve of a known number of *recA* gene extracted from strain B31-A3, and burdens were normalized to 100 ng of total DNA. For the ankles that were applied to histological analysis of Lyme disease-associated joint inflammation (Fig. 4), the analysis was performed as described[37]. The image was scored based on the severity of the inflammation as 0 (no inflammation), 1 (mild inflammation with less than two small foci of infiltration), 2 (moderate inflammation with two or more foci of infiltration), or 3 (severe inflammation with focal and diffuse infiltration covering a large area).

## Crystallization and structure determination

Initial crystallization trials for CspZ-YA and CspZ-YA$_{C187S}$ were performed in polystyrene 96-well sitting-drop crystallization plates with two protein wells per reservoir well (SWISSCI AG, Neuheim, Switzerland). Each reservoir well was manually filled with 50 μl of precipitant solution using sparse-matrix screens JCSG+ and Structure Screen 1&2 from Molecular Dimensions (Newmarket, UK) with a 12-channel pipette (Eppendorf, Hamburg, Germany). Crystallization drops were set up using a Tecan Freedom EVO100 workstation (Tecan Group, Männedorf, Switzerland) by mixing 0.4 μl of protein solution (10 mg/ml in 10 mM Tris (pH 8.0)) with 0.4 μl of precipitant solution. The plates were stored in a temperature-controlled room at 22 °C. Initial crystal hits were observed within 1 to 3 weeks under the following conditions: D7 (0.2 M ammonium sulfate and 30% PEG 4000) for CspZ-YA and C6 (2.0 M ammonium sulfate, 0.1 M HEPES (pH 7.5), and 2% PEG 400) for CspZ-YA$_{C187S}$, both from Structure Screens 1&2. The corresponding crystallization conditions were optimized by varying the quantities of the components in the precipitant solution to obtain crystals suitable for harvesting. Diffraction data for CspZ-YA was collected from crystals grown in 0.2 M ammonium acetate, 0.1 M sodium citrate (pH 6.5), and 30% PEG 4000, but for CspZ-YA$_{C187S}$ crystals grew in 2.2 M ammonium citrate, 0.1 M HEPES (pH 7.5) and 2% PEG 400. Before harvesting and storing the crystals in liquid nitrogen, crystals for CspZ-YA were cryoprotected with 10% glycerol. Diffraction data for CspZ-YA were collected at the Diamond Light Source (Oxfordshire, UK) beamline I03 but the data for CspZ-YA$_{C187S}$ at the MX beamline instrument BL-14.1 at Helmholtz-Zentrum (Berlin, Germany)[72]. Reflections were indexed by XDS and scaled by AIMLESS from the CCP4 suite[73,74]. Initial phases for CspZ-YA and CspZ-YA$_{C187S}$ were obtained by molecular replacement using Phaser[75], with the crystal structure of *B. burgdorferi* CspZ as a search model (PDB ID 4CBE)[68]. After molecular replacement, the protein models were built automatically in BUCCANEER[76]. The crystal structures were improved by manual rebuilding in COOT[77]. Crystallographic refinement was performed using REFMAC5[78]. The structures were validated using Phenix[79]. A summary of the data collection, refinement, and validation statistics for CspZ-YA and CspZ-YA$_{C187S}$ is given in Table S2.

## Protein 3D structure prediction using AlphaFold

AlphaFold v2.0[46] was used to predict the three-dimensional structure for CspZ-YA$_{I183Y}$ as described previously for *B. burgdorferi* PFam12 family proteins[80].

## Molecular dynamics (MD) simulations

MD simulations of *B. burgdorferi* B31 CspZ, CspZ-YA and CspZ-YA$_{C187S}$ (PDB IDs: 9F7I, 9F1V, and 9F21, respectively), as well as the predicted structure of CspZ-YA$_{I183Y}$, were performed using GROMACS[81] version 2021 on the high-performance computing (HPC) infrastructure of Riga Technical University with the CHARMM36[82] force field (version July 2022). All MD simulations were performed under physiologically representative salt conditions (150 mM NaCl). Two sets of simulations were carried out: one at 300 K (default) and the other at 310 K (representing the physiological temperature of 37 °C in mammals). Protein chain termini in the PDB files were capped using PyMOL before system setup in GROMACS. The leap-frog algorithm with a 2 fs time step was used for the integration of the equations of motion. Energy minimization of all systems was performed in ≤100,000 steps. Equilibration involved the generation of an NVT (constant number of particles (N), volume (V), and temperature (T)) ensemble for 100 ps using a V-rescale thermostat, followed by a 100 ps NPT (constant number of particles (N), pressure (P) and temperature (T)) ensemble run using Berendsen barostat with a reference pressure of 1 bar. Position restraints were applied to all backbone atoms with a force constant of 1000 kJ mol$^{-1}$ nm$^{-2}$. Systems simulated at 310 K underwent equilibration at the respective temperature.

Production runs were conducted for 300 ns for each protein at both temperatures. The TIP3P water model[83] was used to solvate all systems. Electrostatic interactions were calculated using the Particle Mesh Ewald algorithm[84] with a real-space cutoff of 1.2 nm. Van der Waals interactions were cut off at 1.2 nm, with the switching potential applied after 1.0 nm. Covalent bonds involving hydrogen atoms were constrained using the LINCS (LINear Constraint Solver) algorithm[85]. During production runs, the V-rescale thermostat was employed for temperature coupling, while isotropic pressure coupling was achieved using the Parinello-Rahman barostat algorithm with a reference pressure of 1 bar. The root mean square deviation (RMSD), the root mean square fluctuation (RMSF), and distance measurements were analyzed using Grace-5.1.25. Trajectories were visualized, and distance measurements were performed using VMD[86].

## Epitope prediction

Prediction of epitope positions for *B. burgdorferi* B31 CspZ, CspZ-YA, and CspZ-YA$_{C187S}$ (PDB IDs: 9F7I, 9F1V, and 9F21, respectively), as well as the predicted structure of CspZ-YA$_{I183Y}$, were based on a matrix of lowest coupling energies (MLCE) and determined using the binding epitope prediction from protein energetics (BEPPE) script[47] (version 1.0.5 installed from https://github.com/colombolab/MLCE) running under default parameters on AMD Ryzen Threadripper 2990 WX 32-Core system with 128 GB RAM and four NVIDIA GeForce RTX 2080 GPUs.

## Surface plasmon resonance (SPR)

Interactions of CspZ-YA with #1139c or #1193c were analyzed by SPR using a Biacore T200 (Cytiva). Ten micrograms of #1139c or #1193c were conjugated to a Sensor Chip Protein G (Cytiva) by flowing each of these IgGs at the flow rate of 10 µl/min, 25 °C through that chip using PBS as the buffer. For quantitative SPR experiments, 10 µL of increasing concentrations (0, 15, 31.25, 62.5, 125, 250, 500 nM) of CspZ-YA were injected into the control cell and the flow cell immobilized with #1139c or #1193c at 10 µl/min, 25 °C. To obtain the kinetic parameters of the interaction, sensogram data were fitted by means of BIAevaluation software version 3.0 (Cytiva), using the one-step biomolecular association reaction model (1:1 Langmuir model), resulting in optimum mathematical fit with the lowest Chi-square values.

## Phagocytosis assays

The phagocytosis assays performed in Fig. S6E have been described previously with modifications[87]. *B. burgdorferi* B31-A3 was labeled with carboxyfluorescein diacetate succinimidyl ester (CFSE, Thermo Fisher Scientific) as described in the vendor's manual. The suspension of spirochetes (10$^7$) in BSK II media without rabbit sera, gelatin, and BSA was incubated with 3.3 µM CFSE at room temperature for 10 min. The normal or heat-inactivated human sera that are treated with antibodies and determined negative for anti-C6 IgGs were incubated with CFSE labeled spirochetes (10$^7$ bacteria) in the presence of #1139c, #1193c, or irrelevant human IgG (Human IgG isotype control, Sigma-Aldrich) at room temperature for 10 min. The spirochete suspension was then mixed with freshly isolated human neutrophils (PMNs) from a blood donor (iQBioscience, Alameda, CA) at the ratio of 25 to 1 and shaking at 37 °C, 50 rpm for 10 min. For each sample, the bacteria-PMNs mixture incubated on ice for 10 min immediately after mixing was included as a control. Phagocytosis was stopped by transferring the bacteria-PMN mixtures to ice-cold fluorescence-activated cell sorting (FACS) buffer (PBS supplemented with 0.5% bovine serum albumin (BSA), 0.01% NaN$_3$, and 0.35 mM EDTA) and stored at 4 °C. Samples continually kept on 4 °C were used as a control. PMNs were then washed and suspended with an ice-cold FACS buffer prior to being applied to a FACSCalibur flow cytometer (Beckton Dickinson). The phagocytosis index of each sample was calculated as mean fluorescence intensity (MFI)×percentage (%) positive cells) at 37 °C minus (MFI×% positive cells) at 4 °C. Each sample were performed in seven replicates in two different events.

## Fluorescence-based thermal shift assays

Ten µM of indicated wild-type and mutant CspZ-YA proteins was applied to a 7500 fast real-time PCR system (Thermo Scientific) with a temperature range of 25–95 °C with an emission wavelength of 623 nm. All reactions were undertaken in 20 µl in 96-well plates using Protein Thermal Shift™ Dye Kit (Thermo Fisher Scientific) at 1:1000 dilutions in PBS. The protein-unfolding concentration (Tm) was extrapolated by obtaining the temperature with maximal positive derivative values of the fluorescence intensity using the 7500 Fast Real-Time PCR System software (Thermo Scientific).

## Accelerated stability study

One µg of untagged CspZ-YA or CspZ-YA$_{C183S}$, or histidine-tagged CspZ-YA or CspZ-YA$_{I183Y}$ was incubated at 4 or 37 °C for 6- or 24-h prior to coating on ELISA plate wells as described[37]. The ELISA plate wells immobilized with untagged or histidine-tagged CspZ-YA before incubation were included as control. After blocking those plate wells by PBS with BSA as described[37], #1139c or #1193c (1 µM), was added to those wells, and the levels of binding between each of these antibodies with CspZ-YA proteins were determined by ELISA as described in the section "ELISAs." Data were expressed as the proportion of #1139c- or #1193c-binding from the ELISA plate wells immobilized with the CspZ-YA proteins incubated at different conditions to those with the control wells.

## Statistical analyses

Significant differences were determined with a Kruskal–Wallis test with the two-stage step-up method of Benjamini, Krieger, and Yekutieli[88], two-tailed Fisher test (for seropositivity in Fig. 3B, H, N)[89], or Spearman analysis (for correlation analysis in Fig. 5B–D and F–H)[90], using GraphPad Prism 9.3.1. A *p* value <0.05 was used to determine significance.

## Reporting summary

Further information on research design is available in the Nature Portfolio Reporting Summary linked to this article.

# Data availability

The coordinates and the structure factors for CspZ-YA and CspZ-YA$_{C187S}$ have been deposited in the Protein Data Bank under accession

codes 9F1V and 9F21, respectively. The data from bactericidal assays, qPCR, histological scoring of joint inflammation, antibody titers, thermoshift assays, accelerated stability studies, circular dichroism, phagocytic indexes generated in this study are provided in Source Data file. Source data are provided with this paper.

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

## Acknowledgements

The authors thank Patricia Rosa and John Leong for providing *B. burgdorferi* strain B31-A3. They also thank Klemen Strle for valuable advice. The authors also thank the Wadsworth Animal Core for assistance with Animal Care and Leslie Eisele and Renjie Song of Wadsworth Biochemistry and Immunology Core for CD spectroscopy, SPR, and flow cytometry, and Susan Wong from Wadsworth Diagnostic Immunology Laboratory for providing human sera. Diffraction data for *B. burgdorferi* CspZ-YA were collected on Diamond Light Source (Oxfordshire, UK) on I03 beamline for beamtime MX35587-1, but for CspZ-YA$_{C187S}$ on beamline BL14.1 at the BESSY II electron storage ring operated by the Helmholtz-Zentrum (Berlin, Germany). This work was supported by NIH grant R01AI181746 (for A.L.M. and Y.L.), R44AI152954 (for J.M., A.L.M., A.W., and Y.L.), R21AI144891 (for J.M., A.L.M., A.W., and Y.L.), R01AI154542 (for X.Y. and U.P.), and the US Department of Defense, Congressionally Directed Medical Research Programs, Grant Number W81XWH-20-1-0913 (Y.C., A.L.M., T.A.N., M.E.B, W.H.C., Y.L. R.T.K., and Z.L.). The funders had no role in study design, data collection, interpretation, or the decision to submit the work for publication.

## Author contributions

K.B. Conceptualization, methodology, validation, formal analysis, investigation, data curation, writing—original draft, writing—review & editing, visualization, supervision, project administration, and funding acquisition. J.M. Conceptualization, methodology, investigation, and data curation. A.L.M. Conceptualization, methodology, investigation, data curation, and writing—review & editing. A.W. Methodology, investigation, and writing—review & editing. Y.-L.C. Methodology and investigation. J.L. Methodology, investigation, and writing—review and editing. Z.L. Methodology, investigation, and writing—review & editing. X.Y. Methodology, investigation, and writing—review & editing. U.S. Methodology, investigation, writing—review & editing. D.T. Methodology and investigation. I.A. Methodology and investigation. M.-E.B. Conceptualization, methodology, validation, investigation, writing—review & editing, supervision, project administration, and funding acquisition. U.P. Methodology, investigation, writing—review & editing, supervision, project administration, and funding acquisition. C.-L.H. Conceptualization, methodology, validation, formal analysis, investigation, data curation, writing— original draft, writing—review & editing, visualization. W.-H.C. Conceptualization, methodology, validation, formal analysis, investigation, data curation, writing—original draft, writing—review & editing, visualization, supervision, project administration, and funding acquisition. Y.-P.L. Conceptualization, methodology, validation, formal analysis, investigation, data curation, writing—original draft, writing— review & editing, visualization, supervision, project administration, and funding acquisition.

## Competing interests

Y.L. is the inventor of US patent application no. US11771750B2 ("Composition and method for generating immunity to *Borrelia burgdorferi*"). The remaining authors declare no competing interests.
