## [Peer Review file · Nature Communications]

Mechanistic insights into the structure-based design of a CspZ-targeting Lyme disease vaccine

Corresponding Author: Dr Yi-Pin Lin

Version 0:

Reviewer comments:

Reviewer #1

(Remarks to the Author)

Overall, the manuscript reports the design and immunological testing of several point mutants of the protein antigen CspZ-YA from *Borrelia burgdorferi*, as potential vaccine candidates against Lyme disease. In comparison with the wildtype antigen, the mutants present mutations selected to block the binding of host complement inhibitor factor H (FH). The rationale behind the design is based on the notion that the protective epitopes are masked in the CspZ-YA-FH complex. By unmasking these epitopes, the antigens will induce better immune protectivity. Indeed, the authors show, in a pilot study, that two mutants (I183Y and C187S) elicit bactericidal antibodies that protect mice from Lyme Disease symptoms and bacterial colonisation. Moreover, this effect is achieved with fewer immunisations with respect to the wildtype antigen. The data are of significance due to the present lack of an effective Lyme Disease vaccine.

The experimental approaches are appropriate to the aim of the study and involve structure-based design starting from the crystal structure of CspZ-YA for subsequent immunological testing in mice. The study starts with the recombinant production of eight mutant antigens that are reduced to seven due to the insolubility of one. The seven antigens are further refined to two (I183Y and C187S), based on the observations that only these two antigens elicited bactericidal antibodies and protective immunity in mice. To understand the molecular basis of improved immunogenicity of the two mutants, the authors then solve the crystal structure of wildtype CspZ-YA alone (previously published structures are in complex with FH) and the C187S mutant, whereas an AlphaFold, in silico model was generated for I183Y. The structural analyses suggest that improved stability may be the key and accordingly, experimental thermal stability studies confirm this and both mutants are more stable than the wildtype antigen.

The structural aspects of the paper could be more thoroughly addressed and often, the analyses are rather superficial in points and conclusions are made without solid supporting data. A key experiment that is missing, that would help fill these gaps in the structural analysis, are molecular dynamics simulations. Indeed, structural biology studies are almost always coupled to computational biology studies, especially when comparing possible, small conformational differences that may occur upon mutation of a single residue. A main question posed in the paper is why the two mutants have improved immunological properties with respect to the wildtype antigen, since the surface epitopes remain unchanged. This latter observation was made, however, using crystal structures that are representative of a single, typically, most stable conformation of a protein and they cannot provide information on related dynamics. The same applies to an AlphaFold model used for one of the mutants. The involvement of MD is also pertinent for the section where the authors investigate antigen stability. The methods adopted in the paper can provide an idea on global stability changes, but they cannot reveal minute changes in an epitope region for example, that is typically made up of several amino acids. Incorporating MD studies into the paper is a simple thing to do given the availability of 3D structure for the antigens of interest and may help ascertain the structural determinants of immunogenicity of the two mutants. That said, the main, great result is that two new protective antigens were designed.

In addition to revision of the English language in general, efforts to improve the style and clarity of the manuscript should be made. In particular, the paper is written in a manner that takes for granted that the reader is an expert on CspZ. For example, this is a structural vaccinology paper but there is not a description of the structural arrangement of the protein in the first place that may only be seen in the figures. In the results section, the authors compare the residues and mention helix names, but they never mention the nomenclature and the number of helices in the first place. A sentence or two would suffice. Moreover, often the results are laid out without specifying which technique was used to obtain that particular result, so the

reader must jump back and forth between the main text and the methods (and even the figure legends) to work out what experiment led to the conclusion that is being discussed. Often extra results are present in the figure legends only. Examples are the thermal stability and surface plasmon resonance experiments that are not mentioned at all in results and are “discovered” in the methods and figures.

It is not clear from the abstract or introduction that the authors solved the crystal structures of wildtype and a mutant protein (high-resolution structure is not informative per se). Starting from line 66, the second half of the abstract should be rewritten. The immunogenicity of the mutants is discussed it is explained where they come from or how they were designed e.g. starting from structural principles. I also don't think the specific findings reported in the paper can be extended to other vaccine antigens as mentioned in the abstract and discussion, although I agree that structural vaccinology is a great tool in tailoring antigens and you in fact, used it successfully.

The last paragraph of the introduction should also be rewritten. Line 137 – you should make it clear that you solved the crystal structures described in the paper. Also “high-resolution structure” is not informative alone (in lines 133 and 137) and crystal or X-ray crystallography should appear somewhere. Once “high resolution” it need not be repeated, and crystal structure or 3D structure is suffice.

I do not understand the phrase in lines 133-135. The way it sounds is that the crystal structure of CspZ-YA can be a model for any antigen, but this is not so. The same applies to lines 139-140. Structural vaccinology is already well established. There are many examples in the literature of how knowledge of the 3D structure can be used to design a better antigen. The work in the manuscript is just another example but it is not a “model”. I think the understanding of the molecular basis of CspZ immunogenicity has been enhanced but it is not a general rule.

There are a few issues to address with regards to the structural data.

1. In the crystal structure solved with PDB code 9F21, the structure validation report reports many rotamer outliers and RSRZ outliers (12% residues is higher than normal). Efforts should be made to improve the structure's geometry. There are also two Ramachandran outliers in both structures that should be fixed or explained in the text. In both validation reports there are also the side chains of 4 residues that are suggested to be flipped. This is a simple thing. Was this done?
2. The phrase stating that atomic coordinates and structure factors were deposited under PDB codes ... is not present. The code is inserted in the text as if the structures have already been released but they have not.
3. The manuscript reports structural comparisons made with the CspZ/SCR6-7 complex (PDB code 9F7I) for which the coordinates have not been released. It is not possible to publish a paper, comparing your data with an unpublished structure that is not available to the scientific community. You should remove this structure from your comparisons or release the atomic coordinates.

Here are additional point-by-point comments:

Introduction:

Line 121 – “by” should be removed to have the correct sense of the phrase. Otherwise, it should be rewritten.

Line 125 – “masked” is a better term to substitute “saturated”

Line 128 – remove “selectively”

Line 129 – rephrase “of the epitopes on this protein's FH binding site” to the epitopes present in the region of CspZ-YA that is bound by FH” or something similar.

Results:

In the first section, Lines 143 onwards, you talk about the Y211A and Y207A mutations, but you need to clarify that these mutations relate to the previously determined structure otherwise its confusing. It is in this section that I would expect at least a brief description of the CspZ-YA fold to set the scene for then talking about specific residues and helices later on.

Lines 145-146 - The PDB codes you used for comparison should be cited together with their references in the text and not just in the legends. In the figure 1 legend that is related to this section, one of the complexes you compare the CspZ-YA structure to is the CspZ/SCR6-7 complex (PDB code 9F7I) that has never been released. You also talk about the loss of an interaction between R206 and E186 but it is not clear what the significance of this one interaction is. What type of bond is it, which atoms are involved? what length is it and what role does it normally have? This is where molecular dynamics may be helpful.

Line 146 – which program was used for structural superpositions- it should be mentioned somewhere in the paper, and the reference cited? the RMSDs written in the figure legend should also be included here in the text. This is an example of data not being reported in text and elsewhere.

Line 154 – 158: The way the substitutions are listed, sounds like you are removing prolines, or polar residues etc. rather than introducing them. Please reword.

Line 245- 247: Were you unable to crystallize the I187Y mutant? (strange if it is more stable than the wild-type protein) and that is why you used AlphaFold? Please add a line to clarify this. You must reference AlphaFold2 when it is mentioned.

Line 264 – Y207, F210 and Y211 are said to form a hydrophobic core but they are on the same side of an amphipathic helix and make a patch rather than a core. In fact, they form hydrophobic interactions with I169 in helix H which is your target for mutation (I183). A better analysis and description of the interactions is required.

In section 3 (i) Line 224, there's a discussion on the hypothesis that the mutants have epitopes distinct from CspZ-YA but it is not clear where and which residues comprise the “surface epitopes” in CspZ-YA in the first place. What are the epitope residue numbers and where is it located on the structure? Fig. 4I should show the surface epitopes but I cannot locate them- I can only see a superposition. Molecular dynamics, perhaps coupled to structure-based computational epitope predictions may have been useful in this section. AlphaFold models or crystal structures cannot provide the full picture on protein dynamics, which is fundamental stimulating an immune response. In this section, immune sera recognition tests show that IgGs from infected parents recognise wildtype and mutant antigens to the same extent. This may mean that the

immunodominant epitope is unaffected, but could there also be less dominant epitopes in the mutant antigens that are not revealed in this assay? Maybe there are slight conformational changes in an epitope, but this cannot be seen without MD.

In section (ii) Line 254, the focus is on stability changes in the mutants due to an increase in the number of interactions induced by the mutated residues but there is no in-depth structural analysis presented on the actual contacts that differ in the wildtype and mutants. In the experimental thermal stability experiments, there is a huge difference in T_m that cannot be explained by the structural analyses provided in section (ii). A water molecule cannot explain this. Once again MD, would provide a more complete picture of the differences in stability and intramolecular interactions.

Section iii – Line 280 – add in the full description “Melting temperature” and put “ T_m ” in brackets. It is not clear which technique the T_m values were determined with. One has to go to the methods and materials to find out that it was a fluorescent-dye based assay. It should already be mentioned here in the results section. Two positions after the decimal point is too much for the sensitivity of a thermal shift assay and the standard deviations should be shown (they are shown in Table S3 but should also be reported in the text).

In this same section, the binding of humanised IgGs to the various antigens is assessed but nothing is mentioned about the method used to do this and no mention of the binding affinities is provided. They are reported in the figures, but something should be mentioned here. It is deduced that the method is SPR only in the methods section.

Line 296- “significantly lower levels of recognition”. Here you should be more scientific e.g. 2-fold difference in binding affinity etc.

Line 297 – SEC-HPLC and not SE-HPLC

Figures:

In general, the graphics programs used to create the structure figures should be cited together with their references. As a general observation, the colours used for the structures are quite muted and sometimes it is difficult to immediately find the focus of the figure. Please indicate the N- and C-termini of the proteins in the figures when visible.

Figure 1A – The significance of the interaction was raised in my point above. If you decide to keep it, please add the bond length.

Figure 1B – some amino acid labels are sticking out of the boxes.

Figure S1 – correct the label on the y axis. It is ellipticity (the “t” is missing).

Table S1 – please show the number of reflections (total and unique) for the high-resolution shell, as has been done for the other parameters.

Please add the $CC1/2$ values to the table and double check the table values as some are different from those reported in the PDB validation reports. The % of reflections taken for the calculation of R_{free} should be mentioned in the legend. It is common practice to cite equations for the calculation of R_{free} and R_{merge} and $CC1/2$ as footnotes.

Figure 4I and J – It is difficult to distinguish between the grey, green and brown ribbons in both panels. In panel I, I cannot see the surface epitopes that are apparently shown (do you deduce no surface epitope changes from the lack of changes in the global fold?). I cannot understand which precise region of the protein hosts the epitope-is it even known? In J, there are dotted lines showing interactions between residues in the background, but they are never discussed in the text, and it is thus unclear the relevance of these interactions (and which residues they are). In the text (line 250) instead, reference to Fig. 4J is made in the context of the solvent inaccessibility of I183 and C187, so it's confusing to show random interactions in the background.

For clarity, in all figures, you should remove stick representation for all residues that are not the focus of the figure.

Figure 5B - the cavity found in CspZ-YA is difficult to appreciate in the figure. Perhaps surface representation would help. Also, an analysis (volume, contributing residues etc.) of this cavity would be useful, using for example servers like CAST-P. The cavity would also be better observed by underlining its absence in the wildtype structure by putting panels 5C and D in the same orientation.

Figures 5C,D and E – Remove sticks from all panels for residues that are not the focus as it looks messy. Which interactions do the dotted lines refer to in the background – they are not informative if the residues aren't labelled or ever discussed.

Figure 6B – Panel B is redundant, and it is sufficient to mention the T_m values in the text and in Table S3. In the legend for Figure 6, Line 1091 – typo “fluoresces” should be “fluorescence”. What is the emission wavelength used for the experiment?

Materials and Methods

Some sections are not detailed enough for a person to repeat them. Depending on the section, no companies are mentioned e.g. after Protein A affinity chromatography, or after the bacterial strains and plasmids. This should be checked throughout.

Lines 419 - There is no information on protein sequences with relative accession codes

Lines 427 -Which growth media and temperatures were used for overexpression?

Lines 431 – Please do not use the first person. A little detail, even the name of the procedure that was used to previously purify CspZ-YA would be useful e.g. “purified by affinity chromatography”.

Line 534- For the primers, please show the 5' and 3' labels.

Line 548 – Crystallization section. What is the volume of the reservoir solution crystallization temperature and growth time? There are many different crystallization plates (1-well, 3-well, flat bottom, round bottom etc.). Please specify. Also please

specify the protein concentrations used and the protein buffer. Which hits produced crystals? The actual condition number and relative screen should be mentioned. (v/v) must be put for the PEGs and glycerol. Which program was used for structure validation? There is no mention of the number of chains in the asymmetric unit (Matthew's coefficient and estimated solvent content) in results or Table S1.

Line 564 – add the reference that corresponds to PDB code 4CBE.

Discussion

The discussion should be rewritten to be more concise and not a simple recap of the results. Furthermore, sometimes conclusions are derived without the proper foundations or data to do this. This is a structural vaccinology (SV) study but a mention of some successful (and famous) examples and relative review articles are not thoroughly presented. Examples of structural modifications are mentioned but a few lines would be useful.

Other points:

Line 303 – the safety issue is also a primary concern in using native antigens.

Line 318 – Change “paired” to “compared.” Lines 317-320. Comparing the structures does not provide evidence supporting that CspZ-YA cannot bind FH. It can hint at this, but only experimental tests can confirm this. Please rephrase.

Lines 320-322 – this phrase is confusing. What do you mean by “potential mechanisms underlining the antigen engineering concept”. Do you mean, a potential strategy to unmask protective epitopes?

There is confusion in the use of the term “Structure-based vaccine design,” which simply means using 3D structure information to engineer a better antigen that can be used in a vaccine. In line 325, the phrase should be changed to read, for example, that 3D structure information may be used to design antigens with improved biochemical properties e.g. stabilities and immunological properties.

Line 330 – you did not provide evidence that this cavity destabilised the protein's conformation. This is something that you can postulate only, without data such as MD simulations.

Line 339 “at one of the first times” is not correct English.

Line 341 – “the other strategy of structure-based vaccine design” makes it sound as if there are a limited number of engineering approaches. Better - “Another structural modification that may be made to improve antigen properties is the manipulation of ...” This paragraph (lines 341 – 358) is all very circumstantial without a proper in-depth analysis of the intramolecular interactions and dynamics between the wildtype and mutant proteins.

Lines 387-388 – I do not believe that the study represents a pipeline. Using structures to design better antigens is already common practice.

References

Some references cite the doi and some do not. Wherever a PDB is mentioned (in text or legends), the paper in which its published should be cited.

Reviewer #2

(Remarks to the Author)

This paper describes a significant, if not novel, and important contribution to the development of vaccines for Lyme disease. They made a number of additional structure modifications to their CspZ-YA vaccine candidate (a mutant of CspZ deficient in FH binding that generates borrellicidal antibodies after immunization, which has been published) and resolved the chemical structure of 2 of the mutants to understand which changes contributed to increased efficacy of the new mutants. They also confirmed that their previously generated monoclonal antibodies (now humanized) known to block the FH-binding of CspZ and promote lysis and opsonophagocytosis of Bb, bind to the new CspZ mutants thus confirming that the epitope that promotes induction of protective antibody is functional.

Strengths of the study: mutations of CspZ-YA stabilized the protective epitopes of the protein (thermostability at physiologic temperature) and enhanced intramolecular interactions between helices H and I, that did not alter the surface epitopes of CspZ-YA, did not change immunogenicity of the molecules and generated two mutants with increased bactericidal activity. Efficacy was established using standard methods by tick transmitted *B. burgdorferi* infection of vaccinates mice. Circular dichroism spectroscopy, crystallization and structure determination, AlphaFold prediction were used to get the structures of the new proteins and surface plasmon resonance was used to interrogate binding of the monoclonal antibody to the new CspZ mutants.

Weaknesses: tick challenge was performed using ticks harboring one strain of Bb (B31 OspC type A) rather than ticks carrying multiple strains of Bb; presence of live Bb was not confirmed from tissues by culture; extremely young mice were used in these experiments: 4 week old pre-adolescent pups rather than adult mice, thus the immune system is not fully developed. All these factors need to be taken into account when making statements of efficacy and need to be included in the discussion. The data can not be compared directly with vaccine efficacy data in the literature as such studies were done using adult mice.

Introduction and references

Both are a bit long.

A few inconsistencies with the literature in Intro:

Line 109: OspA is not only produced by Bb in the tick phase of the cycle (ref 23 is from 1995, please take into account updated literature).

Line 110-111: if there are records of T cell memory to OspA you need to reference this properly. Apply the same judgement to all other references. There's a very large number of references in this paper (80) which is unnecessary, and it is your responsibility to ensure each ref reflects progress in the field and supports what you state.

Lines 120-122: does not make sense.

Results

In all subtitles in this section related to immunization please include the age of the mice. For example, "2. CspZ-YA x and x vaccination of pre-adolescent C3H-HeN mice ...". This is important for the field given that work on OspA and other vaccines for Lyme disease has been traditionally done using adult mice (>7-8 weeks old) and these differences in OspA efficacy are likely due to an immature immune response (you use OspA as a control).

Figure 3. Please include in this figure the data placed in Fig S3 to show the difference between the 3 immunizations (one, 2, 3 doses) because it is important to see that efficacy of the immunogen actually increases with increased vaccine doses.

The arthritis results (3G) should be a different figure. HE results are not enough to define arthritis. However, this is an important figure to the field that shows that unstabilized CspZ-YA induced as much inflammation in the joint as PBS infected control and it was slightly more than OspA (although the differences don't appear to be significant). Make sure to add a statement on this in the discussion.

Figure legends overall: redundant methodology included and extensive description of results reproducing what is described in the text. This can be much simplified.

Figure legends description of immunization procedures: please include the following in the legend "Immunization of pre-adolescent mice with XYZ..." . Make sure to define the age of these mice in all figure legends including supplemental material.

Results Lines 186-188: you can include the fold change or numbers of % bactericidal activity; Lines 207-213: you can say all this in one short sentence; Line 218: since you used HE staining and no markers specific for neutrophils or monocytes it is more accurate to say mononuclear cells infiltrates.

Discussion

Discussion Lines 303-305: rephrase this as there is plenty of evidence that native antigens are effective immunogens as you describe below;

lines 326-340: structure based vaccine design is not new or recent for Borrelia antigens. Please check and reference the work done on the structure of OspA (Koide, Lawson, 2000 and Koide, Luft, 2005) and discuss your data in context.

Line 381: this statement is not true: please check Gingerich 2024 that used a prime-boost immunization scheme using OspA and achieved protection beyond 1 year.

Reviewer #3

(Remarks to the Author)

The manuscript by Brangulis et al. reports on the vetting of a mutated CspZ as a protective immunogen to prevent Lyme borreliosis. The approach used—to immunize with a factor H binding deficient form of CspZ (designated as CspZ-YA) ,and further enhanced by directed mutagenesis, is well conceived and executed. The improved antigenicity and protection provided by various CspZ derivatives is impressive. Overall, the data presented is convincing and mostly easy to navigate. I have a couple of resolvable issues and several minor comments that are intended to assist placing this work in a different contextual framework and clarify the content, respectively. The issues include: (1) The approach is somewhat oversold in its novelty. While unique to Lyme immunization and perhaps bacterial antigens, this strategy (as stated in the Discussion) has been employed with viral antigens; and (2) The seemingly critical loss of factor H binding is mentioned as a key component of the vaccination strategy but further discussion as to why this is a more effective strategy is not expanded. Finally, the Discussion is rough to read in areas because of some awkward verbiage. Some suggestions to address this are listed in the minor comments.

Major Comments:

1. Title. The title does not directly mention CspZ. Given that it is the only molecule tested with this approach, it should be indicated.
2. Abstract. Some of the text will not be easy to navigate when the abstract is free standing (in PubMed, etc.). Please define CspZ-YA or use a different descriptor. Also, altered interaction between helix H and I lacks the context needed for readers to understand this here. A more general description of what was modified by the findings should be indicated instead.
3. Lines 111-112. The reference to issues with the OspA vaccine are not pertinent to vaccine development, but really are more in line with compliance and the efficacy (or the resulting lack of it).
4. Lines 137-140. While the analysis with the mutagenized CspZ reported here is good, the concept that this advances the "molecular basis of modern vaccine strategies" is overstated. As referenced later in the Discussion (lines 333-336), there is a precedent for this approach already. The work here certainly supports this, and the text should instead reflect that sentiment.
5. Fig. 2. It is not clear what the y axis on Fig 2C, E, and G are referring to here. What is the BA50 at a value great than 100 mean? Is it an inverted value? Is it linked to the slopes from Fig. 2B, D, and F? Please clarify. The y axis would appear to log scale? If so, the hatch marks showing this should be indicated.
6. Discussion. Lines 320-322. The concept that the inability of CspZ to no longer bind to factor H makes it an improved

vaccinogen by exposing protective epitopes is intriguing. However, these epitopes would be masked when the native *B. burgdorferi* infects since factor H would be likely be bound to wildtype CspZ. If so, how would the mutagenized CspZ derivatives provide an improved antibody repertoire? Is the concept that native CspZ immunizations fail because they are processed when factor H is bound to them yielding neoantigens that have no protective value? Why would the mutant CspZ, that cannot bind factor H generate antibodies to epitopes that would be pertinent to recognizing CspZ bound by factor H?

7. Lines 361. Not all mammals have a core temperature of 37 degrees Celsius (close but some are warmer). Also, the term "stays consistent" should be reworded as a "normal core temperature".

8. Lines 380-383. The term "constant immunization" in refer to OspA vaccination is not accurate. Instead, repeated vaccination is required to maintain a high titered antibody response for efficacy.

9. Lines 387-388. If the premise that the absence of function shown here for CspZ is critical for new epitopes being exposed and protective antibodies generated, then this statement is not well supported as one would need to know binding partners to effectively develop such a pipeline. Deletion of this sentence is suggested.

Minor Issues:

1. Line 98. The end of this sentence should be reworded for clarity.
2. Lines 180-182. This sentence needs to be fixed. There seem to be some words missing.
3. Lines 182-188. Please define what the term BA50 is referring to. Individuals who do not do this type of analysis may not know this terminology.
4. Lines 190-191. The term "doses" is used here. Aren't these immunizations?
5. Fig. 2C and 2E. I understand that it might be busier, but the x-axis should be labeled for these panels. This same issue is pertinent to Fig. 3 (B and C) and Fig. 6 (C and D) as well.
6. :Lines 192-195. This sentence should be re-written for conciseness.
7. Lines 202-204. This seems implicit and can be re-written to reflect the uninfected control data.
8. Fig. 4. CspZ-YA-C183S is not observed. It this because it is completely superimposable to CspZ-YA?
9. Lines 263-265 and Fig. 5C. F210 is not visible in Fig. 5C. It is observed in Fig. 5D.
10. Lines 338-340. This sentence is oversold in its description and is awkwardly worded.
11. Lines 355-358. This sentence is awkwardly worded.
12. Line 363 end of sentence. Replace "mammal uses" with "mammals".
13. Line 374 end of sentence. "investigation" instead of "investigations"
14. Line 376 second word. Delete the first "the" in this line.
15. Line 379 end of sentence. Would replace "protectivity" with "promotes (or allows) protection".
16. Line 383. The term "sparked off" is slang. Please modify.
17. Line 384. The word "efficacy" should be "efficacious".
18. Line 385. Would rephrase "concept-proof" as "proof of concept".
19. Line 387. The term "inefficacious" could be switched to "nonefficacious".
20. Line 406. Where are BALB/c mice referred to here used? For the monoclonal antibodies?
21. Line 430. The term "protein-derived mutant protein" is not clear.
22. Line 585-603. Where are the phagocytosis assays described here?

Version 1:

Reviewer comments:

Reviewer #1

(Remarks to the Author)

The revised version of the paper is much improved, also due to the integration of new data analyses carried out on the structures. The authors properly addressed all my comments and included a new main figure (Figure 7), illustrating the MD and epitope predictions.

Please add in the MLCE reference (Scarabelli et al., 2010) in Line 263. Since MLCE is based on identifying amino acids pertaining to flexible hot spots, the N- and C-termini are often picked out erroneously as epitopes, since the termini of a protein are often "naturally" flexible. Maybe a phrase should be added to mention this.

The English has been improved but I still find the discussion too long (it is longer than the introduction) and it lacks structure. For example, Lines 348 -351 describe what structural vaccinology can do. Then the subject changes and then in Line 368, we go back to what structural vaccinology can be used for. There are many results that are directly repeated rather than summarising them in the context of the field/state-of-the-art.

Reviewer #2

(Remarks to the Author)

The major issues raised in my previous review have been addressed and i am happy to endorse this paper.

I leave a few minor comments below to be addressed at the discretion of the authors:

- Line 190 – I am not sure OspC type A is the most prevalence in North America. If so please add a recent reference.
Line 203 – Mice don't get Lyme disease. It is more accurate to say "from *B. burgdorferi* infection with .."

Line 227-230 – given that your OspA vac mice have inflammatory scores similar to the control it is prudent to tone down this link between OspA vaccination and inflammation in the joint.

The reference list remains too long for an original research article: as this is not a review article you can use 1 reference for generalist statements (ex line 348).

Reviewer #3

(Remarks to the Author)

The revised manuscript is improved over the initial submission. They have addressed all my major concerns. I only have a few minor comments for the authors consideration.

Minor Issues:

1. Line 87. The term “sparks off” is too conversational. Consider changing this term to “promotes” or something similar.
2. Lines 124. Please consider altering this sentence to read “...vertebrate hosts and, as such, induce CspZ-specific....”.
3. Lines 160-162. This sentence is awkward (particularly the verbiage “rarely being at a distance”). Perhaps this sentiment can be separated into two sentences for clarity. Please consider revising.
4. Lines 207. Put “control” in place of “controlled”.
5. Line 361. Suggest replacing “it” with “from”.
6. Line 404. The end of the sentence (“..., requiring further investigation.”) is redundant to the first part of the sentence and could be omitted.

Point-by-Point Response to Reviewers' Comments

Reviewer #1:

Comment 1:

*Overall, the manuscript reports the design and immunological testing of several point mutants of the protein antigen CspZ-YA from *Borrelia burgdorferi*, as potential vaccine candidates against Lyme disease. In comparison with the wildtype antigen, the mutants present mutations selected to block the binding of host complement inhibitor factor H (FH). The rationale behind the design is based on the notion that the protective epitopes are masked in the CspZ-YA-FH complex. By unmasking these epitopes, the antigens will induce better immune protectivity. Indeed, the authors show, in a pilot study, that two mutants (I183Y and C187S) elicit bactericidal antibodies that protect mice from Lyme Disease symptoms and bacterial colonisation. Moreover, this effect is achieved with fewer immunisations with respect to the wildtype antigen. The data are of significance due to the present lack of an effective Lyme Disease vaccine.*

The experimental approaches are appropriate to the aim of the study and involve structure-based design starting from the crystal structure of CspZ-YA for subsequent immunological testing in mice. The study starts with the recombinant production of eight mutant antigens that are reduced to seven due to the insolubility of one. The seven antigens are further refined to two (I183Y and C187S), based on the observations that only these two antigens elicited bactericidal antibodies and protective immunity in mice. To understand the molecular basis of improved immunogenicity of the two mutants, the authors then solve the crystal structure of wildtype CspZ-YA alone (previously published structures are in complex with FH) and the C187S mutant, whereas an AlphaFold, in silico model was generated for I183Y. The structural analyses suggest that improved stability may be the key and accordingly, experimental thermal stability studies confirm this and both mutants are more stable than the wildtype antigen.

Response:

We appreciate this reviewer summarizing the work and highlighting the significance of this study.

Comment 2:

The structural aspects of the paper could be more thoroughly addressed and often, the analyses are rather superficial in points and conclusions are made without solid supporting data. A key experiment that is missing, that would help fill these gaps in the structural analysis, are molecular dynamics simulations. Indeed, structural biology studies are almost always coupled to computational biology studies, especially when comparing possible, small conformational differences that may occur upon mutation of a single residue. A main question posed in the paper is why the two mutants have improved immunological properties with respect to the wildtype antigen, since the surface epitopes remain unchanged. This latter observation was made, however, using crystal structures that are representative of a single, typically, most stable conformation of a protein and they cannot provide information on related dynamics. The same applies to an AlphaFold model used for one of the mutants. The involvement of MD is also pertinent for the section where the authors investigate antigen stability. The methods adopted in the paper can provide an idea on global stability changes, but they cannot reveal minute changes in an epitope

Point-by-Point Response to Reviewers' Comments

region for example, that is typically made up of several amino acids. Incorporating MD studies into the paper is a simple thing to do given the availability of 3D structure for the antigens of interest and may help ascertain the structural determinants of immunogenicity of the two mutants. That said, the main, great result is that two new protective antigens were designed.

Response:

We thank this reviewer for pointing out the strengths and major weakness (lack of the MD simulation) of this work and have added the data on MD simulation that can now address the possibility of enhanced stability afforded by the two mutant proteins (**Fig. 7 and S1, line 157 to 161 and 290 to 310**).

Comment 3:

In addition to revision of the English language in general, efforts to improve the style and clarity of the manuscript should be made. In particular, the paper is written in a manner that takes for granted that the reader is an expert on CspZ. For example, this is a structural vaccinology paper but there is not a description of the structural arrangement of the protein in the first place that may only be seen in the figures. In the results section, the authors compare the residues and mention helix names, but they never mention the nomenclature and the number of helices in the first place. A sentence or two would suffice. Moreover, often the results are laid out without specifying which technique was used to obtain that particular result, so the reader must jump back and forth between the main text and the methods (and even the figure legends) to work out what experiment led to the conclusion that is being discussed. Often extra results are present in the figure legends only. Examples are the thermal stability and surface plasmon resonance experiments that are not mentioned at all in results and are “discovered” in the methods and figures.

Response:

We have revised the English language generally throughout the manuscript. Additionally, and specifically, we added a sentence or two to clarify the structural arrangement and each technique that was used to obtain these particular results (**lines 145 to 151, 316 to 319, 327 to 328**).

Comment 4:

It is not clear from the abstract or introduction that the authors solved the crystal structures of wildtype and a mutant protein (high-resolution structure is not informative per se). Starting from line 66, the second half of the abstract should be rewritten. The immunogenicity of the mutants is discussed it is explained where they come from or how they were designed e.g. starting from structural principles. I also don't think the specific findings reported in the paper can be extended to other vaccine antigens as mentioned in the abstract and discussion, although I agree that structural vaccinology is a great tool in tailoring antigens and you in fact, used it successfully.

Response:

We revised the abstract and the introduction in the following manner:

Point-by-Point Response to Reviewers' Comments

- 1) Clarified how we came up with the idea of designing these mutant proteins based on structure principals (**lines 61 to 66**)
- 2) Emphasized the significance of this work from testing the concept of structural vaccinology to designing efficacious Lyme disease vaccines. We removed the writing regarding applying the results to other disease systems (**lines 72 to 75, 138 to 141**).

Comment 5:

The last paragraph of the introduction should also be rewritten. Line 137 – you should make it clear that you solved the crystal structures described in the paper. Also “high-resolution structure” is not informative alone (in lines 133 and 137) and crystal or X-ray crystallography should appear somewhere. Once “high resolution” it need not be repeated, and crystal structure or 3D structure is suffice.

Response:

We rewrote the last paragraph of the introduction section (**lines 138 to 141**). Additionally, we have clarified the terminology “high resolution structure” by replacing it with crystal, X-ray structure, or 3D structure as suggested throughout the manuscript (**lines 68, 136, 140, 280**).

Comment 6:

I do not understand the phrase in lines 133-135. The way it sounds is that the crystal structure of CspZ-YA can be a model for any antigen, but this is not so. The same applies to lines 139-140. Structural vaccinology is already well established. There are many examples in the literature of how knowledge of the 3D structure can be used to design a better antigen. The work in the manuscript is just another example but it is not a “model”. I think the understanding of the molecular basis of CspZ immunogenicity has been enhanced but it is not a general rule.

Response:

Lines 133-135 have been rewritten (now, **lines 136 to 137**). Additionally, as outlined above, we have emphasized the significance of this work from testing the concept of structural vaccinology to designing efficacious Lyme disease vaccines. We removed the writing regarding applying the results to other disease systems (**lines 72 to 75, 138 to 141**).

Comment 7:

There are a few issues to address with regards to the structural data. In the crystal structure solved with PDB code 9F21, the structure validation report reports many rotamer outliers and RSRZ outliers (12% residues is higher than normal). Efforts should be made to improve the structure's geometry. There are also two Ramachandran outliers in both structures that should be fixed or explained in the text. In both validation reports there are also the side chains of 4residues that are suggested to be flipped. This is a simple thing. Was this done?

Response:

Both crystal structures reported in the manuscript (PDB 9F21 and 9F1V) were subjected to extensive re-inspection to improve the validation metrics. As a result, the geometry of the structure

Point-by-Point Response to Reviewers' Comments

was improved in both structures which included significant improvements in the clashscore and sidechain outliers. In both structures, the Ramachandran outliers were completely resolved. The sidechains which are suspected to be flipped to improve hydrogen binding and reduce clashes were resolved (**Table S1**).

Comment 8:

The phrase stating that atomic coordinates and structure factors were deposited under PDB codes ...is not present. The code is inserted in the text as if the structures have already been released but they have not.

Response:

We added the section "Data availability" according to the journal guidelines, as suggested by the reviewer (**lines 716 to 717**).

Comment 9:

The manuscript reports structural comparisons made with the CspZ/SCR6-7 complex (PDB code 9F7I) for which the coordinates have not been released. It is not possible to publish a paper, comparing your data with an unpublished structure that is not available to the scientific community. You should remove this structure from your comparisons or release the atomic coordinates.

Response:

The crystal structure of the CspZ-SCR6-7 complex (PDB ID 9F7I) has now been released.

Comment 10:

Here are additional point-by-point comments:

Introduction:

Line 121 – "by" should be removed to have the correct sense of the phrase. Otherwise, it should be rewritten

Response:

"By" has been removed.

Comment 11:

Line 125 – "masked" is a better term to substitute "saturated"

Response:

This has been corrected (**line 129**).

Comment 12:

Line 128 – remove "selectively"

Response:

Point-by-Point Response to Reviewers' Comments

“Selectively” has been removed.

Comment 13:

Line 129 – rephrase “of the epitopes on this protein’s FH binding site” to the epitopes present in the region of CspZ-YA that is bound by FH” or something similar.

Response:

This has been rephrased (**lines 132**).

Comment 14:

In the first section, Lines 143 onwards, you talk about the Y211A and Y207A mutations, but you need to clarify that these mutations relate to the previously determined structure otherwise its confusing. It is in this section that I would expect at least a brief description of the CspZ-YA fold to set the scene for then talking about specific residues and helices later on.

Response:

Please refer to our response to comment 3 from the general comments of this reviewer. The description of the CspZ-YA designation was added in parentheses. Also, a brief description of the overall fold was added in this section as suggested by the reviewer (**lines 147 to 151**).

Comment 15:

Lines 145-146 - The PDB codes you used for comparison should be cited together with their references in the text and not just in the legends. In the figure 1 legend that is related to this section, one of the complexes you compare the CspZ-YA structure to is the CspZ/SCR6-7 complex (PDB code 9F7I) that has never been released. You also talk about the loss of an interaction between R206 and E186 but it is not clear what the significance of this one interaction is. What type of bond is it, which atoms are involved? what length is it and what role does it normally have? This is where molecular dynamics may be helpful.

Response:

The PDB codes together with their references were added in the relevant section as suggested by the reviewer. The crystal structure of the CspZ-FH complex with the PDB ID 9F7I has been released now (1).

More details were added about the interaction between residues R206 and E186 (**lines 153 to 157**). We have used MD simulations to generate the information for the interaction between residues R206 and E186, specifically to examine the distance between both side chains over time in wild-type CspZ, CspZ-YA, CspZ-YA_{C187S} and CspZ-YA_{I183Y} proteins (**Fig. S1, lines 157 to 162**).

Comment 16:

Line 146 – which program was used for structural superpositions- it should be mentioned somewhere in the paper, and the reference cited? the RMSDs written in the figure legend should

Point-by-Point Response to Reviewers' Comments

also be included here in the text. This is an example of data not being reported in text and elsewhere.

Response:

We have added the name of the software (i.e., CCP4MG) used and referenced it accordingly (**line 148**). The RMSD values were added to the text as suggested (**line 149 to 150**).

Comment 17:

Line 154 – 158: The way the substitutions are listed, sounds like you are removing prolines, or polar residues etc. rather than introducing them. Please reword.

Response:

The term “substituting” has been replaced by “mutating” to avoid any confusion (**line 164**).

Comment 18:

Line 245- 247: Were you unable to crystallize the I187Y mutant? (strange if it is more stable than the wild-type protein) and that is why you used Alphafold? Please add a line to clarify this. You must reference Alphafold2 when it is mentioned.

Response:

Although the I183Y mutant is more stable than the wild-type protein (CspZ-YA), we were unable to crystallize it. Therefore, we used Alphafold to predict its structure. We have clarified this and a reference to Alphafold 2 (**lines 255 to 256**).

Comment 19:

Line 264 – Y207, F210 and Y211 are said to form a hydrophobic core but they are on the same side of an amphipathic helix and make a patch rather than a core. In fact, they form hydrophobic interactions with I169 in helix H which is your target for mutation (I183). A better analysis and description of the interactions is required.

Response:

The description of the hydrophobic core was revised to highlight the location of specific residues in the hydrophobic core (**line 280 to 282**).

Comment 20:

In section 3 (i) Line 224, there's a discussion on the hypothesis that the mutants have epitopes distinct from CspZ-YA but it is not clear where and which residues comprise the “surface epitopes” in CspZ-YA in the first place. What are the epitope residue numbers and where is it located on the structure? Fig.4I should show the surface epitopes but I cannot locate them- I can only see a superposition. Molecular dynamics, perhaps coupled to structure-based computational epitope predictions may have been useful in this section. Alpha fold models or crystal structures cannot provide the full picture on protein dynamics, which is fundamental stimulating an immune response. In this section, immune sera recognition tests show that IgGs from infected parents

Point-by-Point Response to Reviewers' Comments

recognise wildtype and mutant antigens to the same extent. This may mean that the immunodominant epitope is unaffected, but could there also be less dominant epitopes in the mutant antigens that are not revealed in this assay? Maybe there are slight conformational changes in an epitope, but this cannot be seen without MD.

Response:

We agree with this reviewer that comparing/superimposing the structures from X-ray crystallography or AlphaFold models is unable to provide the full picture for epitope comparison, especially if the epitopes impacted are less dominant in the antigens. Therefore, to locate the epitopes and specify the involved residues, we introduced putative immunogenic epitope prediction for CspZ-YA, CspZ-YA_{C187S} and CspZ-YA_{I183Y} based on Matrix of Lowest Coupling Energies (MLCE) calculations performed using Binding Epitope Prediction from Protein Energetics (BEPPE). Through MLCE, we were able to identify two patches of epitopes that are present in all three CspZ-YA proteins (i.e., CspZ-YA, CspZ-YA_{C187S}, and CspZ-YA_{I183Y}), further strengthening the conclusion that mutagenesis of C187S or I183Y does not lead to the changes of epitopes. This information has been added to the results section (**Fig. S5 and lines 262 to 266**).

Comment 21:

In section (ii) Line 254, the focus is on stability changes in the mutants due to an increase in the number of interactions induced by the mutated residues but there is no in depth structural analysis presented on the actual contacts that differ in the wildtype and mutants. In the experimental thermal stability experiments, there is a huge difference in T_m that cannot be explained by the structural analyses provided in section (ii). A water molecule cannot explain this. Once again MD, would provide a more complete picture of the differences in stability and intramolecular interactions.

Response:

MD simulations were introduced to explain the observed differences in the stability between wild-type CspZ, CspZ-YA, CspZ-YA_{C187S} and CspZ-YA_{I183Y} (**Fig. 7, lines 290 to 310**).

Comment 22:

Section iii – Line 280 – add in the full description “Melting temperature” and put “T_m” in brackets. It is not clear which technique the T_m values were determined with. One has to go to the methods and materials to find out that it was a fluorescent-dye based assay. It should already be mentioned here in the results section. Two positions after the decimal point is too much for the sensitivity of a thermal shift assay and the standard deviations should be shown (they are shown in Table S3 but should also be reported in the text).

Response:

We have included a brief description of how we obtained the T_m in the result section, added the description of melting temperature and placed T_m in brackets (**lines 316 to 319**). Additionally, we now added one position after the decimal point and included the T_m values and their standard deviation in the results section as suggested (**lines 319 and 320 and Table S3**).

Point-by-Point Response to Reviewers' Comments

Comment 23:

In this same section, the binding of humanised IgGs to the various antigens is assessed but nothing is mentioned about the method used to do this and no mention of the binding affinities is provided. They are reported in the figures, but something should be mentioned here. It is deduced that the method is SPR only in the methods section.

Response:

We have included a brief description of how we obtained the binding affinities for humanized IgGs and CspZ-YA (**lines 327 to 328**).

Comment 24:

Line 296- “significantly lower levels of recognition”. Here you should be more scientific e.g. 2-fold difference in binding affinity etc.

Response:

We replace “significantly lower levels of recognition” with the fold differences in binding affinity (**line 336**).

Comment 25:

Line 297 – SEC-HPLC and not SE-HPLC

Response:

This has been corrected (**line 337**).

Comment 26:

Figures:

In general, the graphics programs used to create the structure figures should be cited together with their references. As a general observation, the colours used for the structures are quite muted and sometimes it is difficult to immediately find the focus of the figure. Please indicate the N- and C-termini of the proteins in the figures when visible.

Response:

The use of a graphics program (i.e., CCPMG) was cited and referenced in the text as suggested by the reviewer (**line 148**). The colors for the structures, for example, in Figure 1A, were made more vivid and expressive (**Fig. 1**). The N- and C-terminus, for example, in Figure 1A were indicated as suggested by the reviewer.

Comment 27:

Figure 1A – The significance of the interaction was raised in my point above. If you decide to keep it, please add the bond the length.

Response:

Point-by-Point Response to Reviewers' Comments

The length of the bond between the guanidine group of R206 and the carboxylate of E186 was added in Fig. 1A as suggested by the reviewer (**Fig. 1A**).

Comment 28:

Figure 1B – some amino acid labels are sticking out of the boxes.

Response:

We have fixed the amino acid labels to ensure they are in the boxes (**Fig. 1B**).

Comment 29:

Figure S1 – correct the label on the y axis. It is ellipticity (the “t” is missing).

Response:

We have fixed that typo (**Fig. S1**).

Comment 30:

Table S1 – please show the number of reflections (total and unique) for the high-resolution shell, as has been done for the other parameters.

Please add the CC1/2 values to the table and double check the table values as some are different from those reported in the PDB validation reports. The % of reflections taken for the calculation of Rfree should be mentioned in the legend. It is common practice to cite equations for the calculation of Rfree and Rmerge and CC1/2 as footnotes.

Response:

We added the number of reflections for the high-resolution shell and the CC1/2 values in Table S1. Likewise, we added the information about the % of reflections used for Rfree calculation and added the calculation for Rfree, Rmerge, and CC1/2 in footnotes.

Comment 31:

Figure 4I and J – It is difficult to distinguish between the grey, green and brown ribbons in both panels. In panel I, I cannot see the surface epitopes that are apparently shown (do you deduce no surface epitope changes from the lack of changes in the global fold?). I cannot understand which precise region of the protein hosts the epitope-is it even known? In J, there are dotted lines showing interactions between residues in the background, but they are never discussed in the text, and it isthus unclear the relevance of these interactions (and which residues they are). In the text (line 250) instead, reference to Fig. 4J is made in the context of the solvent inaccessibility of I183 and C187, so it's confusing to show random interactions in the background. For clarity, in all figures, you should remove stick representation for all residues that are not the focus of the figure.

Response:

Point-by-Point Response to Reviewers' Comments

As this reviewer mentioned, yes, we are trying to overlay the three structures (CspZ-YA, CspZ-YA_{C187S}, and CspZ-YA_{I183Y}) to make the point that the global structure does not change after the amino acids are mutated, and, thus, the surface epitopes likely are not impacted. However, we agree with this reviewer that an X-ray crystal structure only represents single, typically, most stable conformations of a protein and cannot provide information on related dynamics (see comment 2 of this reviewer). Therefore, the X-ray crystal structure may not be sufficient to support the conclusion of no changes *to the surface epitopes*. *We thus included data of epitope prediction based on matrix of lowest coupling energies (MLCE) as suggested by the reviewer (Fig. S5, lines 262 to 266).*

The dotted lines and stick representation are not relevant to the conclusion we would like to make so they are removed (**Fig. 5I and J**).

Comment 32:

Figure 5B - the cavity found in CspZ-YA is difficult to appreciate in the figure. Perhaps surface representation would help. Also, an analysis (volume, contributing residues etc.) of this cavity would be useful, using for example servers like CAST-P. The cavity would also be better observed by underlining its absence in the wildtype structure by putting panels 5C and D in the same orientation.

Response:

In Figures 5A and 5B, the intent was not to illustrate the hydrophobic cavity as shown in Figures 5 C-E, but to show the electron density around C187 and S187 to notice any differences between both structures. We have clarified this intention (**lines 273 to 275**).

Comment 33:

Figures 5C,D and E – Remove sticks from all panels for residues that are not the focus as it looks messy. Which interactions do the dotted lines refer to in the background – they are not informative if the residues aren't labelled or ever discussed.

Response:

We have removed the dotted lines and stick representations as they are not relevant to this study (**Fig. 6C, D, and E**).

Comment 34:

Figure 6B – Panel B is redundant, and it is sufficient to mention the T_m values in the text and in Table S3. In the legend for Figure 6, Line 1091 – typo “fluoresces” should be “fluorescence”. What is the emission wavelength used for the experiment?

Response:

Figure 6B has been removed as suggested. Additionally, the typo (fluorescence) was removed when the figure legends were simplified (see the response to comment 10 of reviewer 2). The

Point-by-Point Response to Reviewers' Comments

emission wavelength was added to the material and methods section as well as to the legend of Figure 6 (**lines 691 to 692 and 1247**)

Comment 35:

Materials and Methods

Some sections are not detailed enough for a person to repeat them. Depending on the section, no companies are mentioned e.g. after Protein A affinity chromatography, or after the bacterial strain and plasmids. This should be checked throughout.

Response:

We have checked throughout the materials and method section to make sure that the description is detailed enough for a person to repeat the studies. The information added includes the companies that the materials were ordered from (**lines 464, 465 to 468, 472, 482 to 483**).

Comment 36:

Lines 419 - There is no information on protein sequences with relative accession codes

Response:

The accession codes of the protein sequences have been added in **Table S4**.

Comment 37:

Lines 427 -Which growth media and temperatures were used for overexpression?

Response:

This information has been added (**line 463**).

Comment 38:

Lines 431 – Please do not use the first person. A little detail, even the name of the procedure that was used to previously purify CspZ-YA would be useful e.g. “purified by affinity chromatography”.

Response:

We rewrote this sentence to avoid using the first person and added details of the procedure, including the name of the procedure to purify CspZ-YA (**lines 465 to 468**).

Comment 39:

Line 534- For the primers, please show the 5' and 3' labels.

Response:

We added the 5' and 3' labels for the primers (**line 578 to 579**).

Comment 40:

Line 548 – Crystallization section. What is the volume of the reservoir solution, crystallization temperature, and growth time? There are many different crystallization plates (1-well, 3-well,

Point-by-Point Response to Reviewers' Comments

flatbottom, round bottom etc.). Please specify. Also please specify the protein concentrations used and the protein buffer. Which hits produced crystals? The actual condition number and relative screen should be mentioned. (v/v) must be put for the PEGs and glycerol. Which program was used for structure validation? There is no mention of the number of chains in the asymmetric unit (Matthew's coefficient and estimated solvent content) in results or Table S1.

Response:

We introduced the information suggested by the reviewer in the Methods section "Crystallization and structure determination". This information includes: details of the crystallization plates; protein concentration and buffer; volume of reservoir solution; crystallization condition numbers and the corresponding screening kit; crystal growth time and temperature; and software for structure validation (**lines 589 to 601**). Matthews coefficient and solvent content of the crystal were added in Table S1.

Comment 41:

Line 564 – add the reference that corresponds to PDB code 4CBE.

Response:

That reference has been added (**line 610**).

Comment 42:

Discussion

The discussion should be rewritten to be more concise and not a simple recap of the results. Furthermore, sometimes conclusions are derived without the proper foundations or data to do this. This is a structural vaccinology (SV) study but a mention of some successful (and famous) examples and relative review articles are not thoroughly presented. Examples of structural modifications are mentioned but a few lines would be useful.

Response:

We rewrote the discussion to simplify the concept we would like to discuss and deleted some sentences to prevent redundancy (the first paragraph of the discussion; **lines 343 to 344**, as well as **lines 379 to 382** and **418 to 420**). Additionally, we have included several relevant review articles and successful (and renown) examples of using this approach to develop the vaccines (**lines 371 to 374**, **348 to 352**).

Comment 43:

Other points:

Line 303 – the safety issue is also a primary concern in using native antigens.

Response:

We have added the description of native antigens and their potential safety issues in the discussion section (**lines 345**).

Point-by-Point Response to Reviewers' Comments

Comment 44:

Line 318 – Change “paired” to “compared.” Lines 317-320. Comparing the structures does not provide evidence supporting that CspZ-YA cannot bind FH. It can hint at this, but only experimental tests can confirm this. Please rephrase.

Response:

We have rephrased this as suggested (**lines 368 and 370**).

Comment 45:

Lines 320-322 – this phrase is confusing. What do you mean by “potential mechanisms underlining the antigen engineering concept”. Do you mean, a potential strategy to unmask protective epitopes?

Response:

What we meant is the potential mechanisms allowing the strategy of unmasking protective epitopes to be capable of promoting immunogenicity. This has been clarified (**lines 364 to 365**).

Comment 46:

There is confusion in the use of the term “Structure-based vaccine design,” which simply means using 3D structure information to engineer a better antigen that can be used in a vaccine. In line 325, the phrase should be changed to read, for example, that 3D structure information may be used to design antigens with improved biochemical properties e.g. stabilities and immunological properties.

Response:

The phrase identified in this comment has been changed as suggested (**lines 348 to 349**).

Comment 47:

Line 330 – you did not provide evidence that this cavity destabilised the protein’s conformation. This is something that you can postulate only, without data such as MD simulations.

Response:

We have performed MD simulations, showing the destabilization of the protein’s conformation in the cavity of CspZ-YA. This information has been added to the manuscript and figures (**lines 157 to 162, Fig. S1**).

Comment 48:

Line 339 “at one of the first times” is not correct English.

Response:

That sentence has been removed

Comment 49:

Point-by-Point Response to Reviewers' Comments

Line 341 – “the other strategy of structure-based vaccine design” makes it sound as if there are a limited number of engineering approaches. Better - “Another structural modification that may be made to improve antigen properties is the manipulation of ...” This paragraph (lines 341 – 358) is all very circumstantial without a proper in-depth analysis of the intramolecular interactions and dynamics between the wildtype and mutant proteins.

Response:

We rewrote the sentence in line 341 as suggested by the reviewer (**lines 383 to 384**).

Comment 50:

Lines 387-388 – I do not believe that the study represents a pipeline. Using structures to design better antigens is already common practice.

Response:

The indicated sentence has been removed.

Comment 51:

References

Some references cite the doi and some do not. Wherever a PDB is mentioned (in text or legends), the paper in which its published should be cited.

Response:

The doi is not required for the style of reference in Nature Communication. We thus have removed all doi information. For PDB citations, we have added the citations associated with the PDB throughout the manuscript. Please also see the response to the comment 40 of the reviewer 1.

Reviewer #2:

Comment 1:

This paper describes a significant, if not novel, and important contribution to the development of vaccines for Lyme disease. They made a number of additional structure modifications to their CspZ-YA vaccine candidate (a mutant of CspZ deficient in FH binding that generates borrellicidal antibodies after immunization, which has been published) and resolved the chemical structure of 2 of the mutants to understand which changes contributed to increased efficacy of the new mutants. They also confirmed that their previously generated monoclonal antibodies (now humanized) known to block the FH-binding of CspZ and promote lysis and opsonophagocytosis of Bb, bind to the new CspZ mutants thus confirming that the epitope that promotes induction of protective antibody is functional.

Strengths of the study: mutations of CspZ-YA stabilized the protective epitopes of the protein (thermostability at physiologic temperature) and enhanced intramolecular interactions between helices H and I, that did not alter the surface epitopes of CspZ-YA, did not change immunogenicity of the molecules and generated two mutants with increased bactericidal activity. Efficacy was

Point-by-Point Response to Reviewers' Comments

established using standard methods by tick transmitted B. burgdorferi infection of vaccinates mice. Circular dichroism spectroscopy, crystallization and structure determination, Alphafold prediction were used to get the structures of the new proteins and surface plasmon resonance was used to interrogate binding of the monoclonal antibody to the new CspZ mutants.

Response:

We appreciate this reviewer summarizing the work and pointing out the strength of this study.

Comment 2:

Weaknesses: tick challenge was performed using ticks harboring one strain of Bb (B31 OspC type A) rather than ticks carrying multiple strains of Bb; presence of live Bb was not confirmed from tissues by culture; extremely young mice were used in these experiments: 4 week old pre-adolescent pups rather than adult mice, thus the immune system is not fully developed. All these factors need to be taken into account when making statements of efficacy and need to be included in the discussion. The data cannot be compared directly with vaccine efficacy data in the literature as such studies were done using adult mice.

Response:

We have addressed each of these weaknesses in the response to the reviewer's comments.

Comment 3:

Introduction and references

Both are a bit long.

A few inconsistencies with the literature in Intro:

Response:

We have shortened the introduction and ensured the literature in the introduction cited correctly (please also see below comments).

Comment 4:

Line 109: OspA is not only produced by Bb in the tick phase of the cycle (ref 23 is from 1995, please take into account updated literature).

Response:

We rewrote this to reflect the updated literature that these vaccines target a Lyme borreliae protein, OspA, that is produced abundantly when bacteria are in ticks. However, OspA production is reduced after bacteria are transmitted to mammalian hosts, making it more difficult for mammalian hosts to produce a significant memory immune response against OspA (**lines 108 to 111**).

Comment 5:

Line 110-111: if there are records of T cell memory to OspA, you need to reference this properly. Apply the same judgement to all other references. There's a very large number of references in

Point-by-Point Response to Reviewers' Comments

this paper (80) which is unnecessary, and it is your responsibility to ensure each ref reflects progress in the field and supports what you state.

Response:

As there is insufficient evidence to support whether T cell memory to OspA through vaccination is achieved, we have removed the relevant sentence regarding the memory immune response. Also, reference (80) is no longer in lines 110 to 111, to better apply the statement to all references. Instead, we added the reference supporting the statement of the need for multiple immunizations to achieve protection with OspA-based vaccines (2). Additionally, some OspA-based vaccines have been developed to reduce the required booster frequency for protection by either using different inoculation routes or conjugating the antigens with immune-stimulating agents (i.e., nanoparticles (3, 4)). Therefore, we have also reframed this section in the text (**lines 108 to 114**).

Comment 6:

Lines 120-122: does not make sense.

Response:

We rewrote this too as “CspZ is only produced after Lyme borreliæ invade vertebrate hosts, likely to have the capacity to induce CspZ-specific immune responses after natural infection. These observations underscore the potential of employing this protein as an attractive Lyme disease vaccine candidate.” (**lines 123 to 125**).

Comment 7:

In all subtitles in this section related to immunization please include the age of the mice. For example, 2. CspZ-YA x and x vaccination of pre-adolescent C3H-HeN mice ...”. This is important for the field given that work on OspA and other vaccines for Lyme disease has been traditionally done using adult mice (>7-8 weeks old) and these differences in OspA efficacy are likely due to an immature immune response (you use OspA as a control).

Response:

We have revised the subtitle in the results section to reflect that our work was done using pre-adolescent C3H-HeN mice (**lines 179**) so the readers would understand the implications of our results regarding the age-mediated differences when compared with others' results.

Comment 8:

Figure 3. Please include in this figure the data placed in Fig S3 to show the difference between the 3 immunizations (one, 2, 3 doses) because it is important to see that efficacy of the immunogen actually increases with increased vaccine doses.

Response:

We have moved the results from Fig. S3 to Fig. 3 (**Fig. 3A to F and M to R**).

Comment 9:

Point-by-Point Response to Reviewers' Comments

The arthritis results (3G) should be a different figure. HE results are not enough to define arthritis. However, this is an important figure to the field that shows that unstabilized CspZ-YA induced as much inflammation in the joint as PBS infected control and it was slightly more than OspA (although the differences don't appear to be significant). Make sure to add a statement on this in the discussion.

Response:

We have moved the histopathology results to Fig. 4 (**Fig. 4**). We replaced the term “arthritis” with “joint inflammation” when describing the histopathology results (**lines 227 to 228**). We have also added one sentence to mention that two immunizations with OspA would prevent one out of five mice from developing joint inflammation while all five mice immunized with CspZ-YA twice had joint inflammation (**lines 229 to 230**).

Comment 10:

Figure legends overall: redundant methodology included and extensive description of results reproducing what is described in the text. This can be much simplified.

Response:

We have simplified the figure legends throughout (**lines 1140 to 1142, 1144 to 1149, 1160 to 1163, 1183 to 1186, 1245 to 1247**).

Comment 11:

Figure legends description of immunization procedures: please include the following in the legend “Immunization of pre-adolescent mice with XYZ...” . Make sure to define the age of these mice in all figure legends including supplemental material.

Response:

We have added the suggested description in the figure legends and also added the age of these mice in all figure legends (**lines 1141 and 1160, and the line 89 in supplementary information**).

Comment 12:

Results Lines 186-188: you can include the fold change or numbers of % bactericidal activity; Lines 207-213: you can say all this in one short sentence; Line 218: since you used HE staining and no markers specific for neutrophils or monocytes it is more accurate to say mononuclear cells infiltrates.

Response:

We have included the fold changes for bactericidal assays (**lines 197 and 199**), simplified lines 207-213 (now **lines 213 to 217**). For HE staining results, we have also replaced the description of neutrophils or monocytes with the term ‘mononuclear cells’ (**line 226**).

Comment 13:

Discussion

Point-by-Point Response to Reviewers' Comments

Discussion Lines 303-305: rephrase this as there is plenty of evidence that native antigens are effective immunogens as you describe below;

Response:

We have rephrased this sentence to “Although many native microbial antigens are effective as immunogens, some of these antigens present challenges to be developed as vaccines.” (**lines 343 to 344**).

Comment 14:

lines 326-340: structure based vaccine design is not new or recent for Borrelia antigens. Please check and reference the work done on the structure of OspA (Koide, Lawson, 2000 and Koide, Luft, 2005) and discuss your data in context.

Response:

We agree with this reviewer (and reviewer 1) that structural-based vaccine design (or structural vaccinology) has been tested in some microbial antigens, including *Borrelia* antigens. We thus rephrased our statement throughout the manuscript (see comment 4 to reviewer 1). Specifically, we have added the previous work done by Koide et al. 2000 and Koide et al. 2005 in the discussion section (**line 352**).

Comment 15:

Line 381: this statement is not true: please check Gingerich 2024 that used a prime-boost immunization scheme using OspA and achieved protection beyond 1 year.

Response:

The statement in line 381 has been removed. We also specified the efforts of Gingerich 2024 in the introduction section (**lines 112 to 114**).

Reviewer #3:

Comment 1:

The manuscript by Brangulis et al. reports on the vetting of a mutated CspZ as a protective immunogen to prevent Lyme borreliosis. The approach used—to immunize with a factor H binding deficient form of CspZ (designated as CspZ-YA), and further enhanced by directed mutagenesis, is well conceived and executed. The improved antigenicity and protection provided by various CspZ derivatives is impressive. Overall, the data presented is convincing and mostly easy to navigate. I have a couple of resolvable issues and several minor comments that are intended to assist placing this work in a different contextual framework and clarify the content, respectively. The issues include: (1) The approach is somewhat oversold in its novelty. While unique to Lyme immunization and perhaps bacterial antigens, this strategy (as stated in the Discussion) has been employed with viral antigens; and (2) The seemingly critical loss of factor H binding is mentioned as a key component of the vaccination strategy but further discussion as to why this is a more effective strategy is not expanded. Finally, the Discussion is rough to read in areas because of some awkward verbiage. Some suggestions to address this are listed in the minor comments.

Point-by-Point Response to Reviewers' Comments

Response:

We appreciate this reviewer's summary of the work and pointing out its strengths and weaknesses of this study. Our responses to specific comments are below.

Comment 2:

Major Comments:

1. Title. The title does not directly mention CspZ. Given that it is the only molecule tested with this approach, it should be indicated.

Response:

We have revised the title to "Mechanistic insights into structure-based design of a CspZ-targeting Lyme disease vaccine" (**line 1**).

Comment 3:

2. Abstract. Some of the text will not be easy to navigate when the abstract is free standing (in PubMed, etc.). Please define CspZ-YA or use a different descriptor. Also, altered interaction between helix H and I lacks the context needed for readers to understand this here. A more general description of what was modified by the findings should be indicated instead.

Response:

We revised the abstract to ensure the terminology would be understandable when the abstract is free-standing. The descriptions of helix H and I have been removed from the abstract section.

Comment 4:

3. Lines 111-112. The reference to issues with the OspA vaccine are not pertinent to vaccine development, but really are more in line with compliance and the efficacy (or the resulting lack of it).

Response:

That sentence has been removed and rewritten as suggested by reviewer 2.

Comment 5:

4. Lines 137-140. While the analysis with the mutagenized CspZ reported here is good, the concept that this advances the "molecular basis of modern vaccine strategies" is overstated. As referenced later in the Discussion (lines 333-336), there is a precedent for this approach already. The work here certainly supports this, and the text should instead reflect that sentiment.

Response:

Those sentences in the introduction and discussion have been rewritten as suggested by reviewer 1 (**lines 138 to 141 and 418 to 420**).

Comment 6:

Point-by-Point Response to Reviewers' Comments

5. Fig. 2. It is not clear what the y axis on Fig 2C, E, and G are referring to here. What is the BA₅₀ at a value great than 100 mean? Is it an inverted value? Is it linked to the slopes from Fig. 2B, D, and F? Please clarify. The y axis would appear to log scale? If so, the hatch marks showing this should be indicated.

Response:

As the dilution of the sera increases, the survivability of the bacteria decreases provided the sera can kill bacteria. To quantitatively compare the ability of sera to kill bacteria, the BA₅₀ value was used to indicate the dilution that kills 50% of *B. burgdorferi* cells. A BA₅₀ value greater than 100 would indicate that the respective sera have the capability to kill 50% of *B. burgdorferi* at a serum dilution of 1: 100. Basically, different dilutions of the sera are tested with *B. burgdorferi* to determine the percentage of surviving bacteria (see **lines 563 to 571**). The plots with dilution rates (using log₂ scale) compared to percent survival (Fig. 2B, D, and E) were then fitted to obtain the BA₅₀ values. These experiments were done using the sera from five mice per group. Thus, these BA₅₀ values derived from sera of mice immunized with each of the indicated immunization frequencies and antigens were plotted and compared in Fig. 2C, E, and G.

We have added a few sentences at the beginning of the description for these experiments to explain how the bactericidal assays were done in the result section (**lines 191 to 193**). We also replace “% borreliacidal dilution of each serum sample” by “BA₅₀ values” in Fig. 2C, E, G, and in the respective figure legends to clarify what we are showing (**Fig 2C, E, and G, and line 1152**).

Comment 7:

6. Discussion. Lines 320-322. The concept that the inability of CspZ to no longer bind to factor H makes it an improved vaccinogen by exposing protective epitopes is intriguing. However, these epitopes would be masked when the native *B. burgdorferi* infects since factor H would be likely be bound to wildtype CspZ. If so, how would the mutagenized CspZ derivatives provide an improved antibody repertoire? Is the concept that native CspZ immunizations fail because they are processed when factor H is bound to them yielding neoantigens that have no protective value? Why would the mutant CspZ, that cannot bind factor H generate antibodies to epitopes that would be pertinent to recognizing CspZ bound by factor H?

Response:

The concept of the unmasking strategy in vaccinology is that the repertoire of antibodies triggered by the immunization of unmasked antigens contains the protective antibody that recognizes the epitopes of native antigens during infection but before those native antigens bind to the binding partner. In the case of CspZ and factor H, it is possible that the protective antibodies triggered by vaccination with CspZ-YA are present in the bloodstream. These antibodies thus can bind to the native antigen, CspZ, on *B. burgdorferi* before that native antigen binds to factor H immediately after *B. burgdorferi* invades the hosts. It is also possible that protective antibodies with higher affinity to CspZ (lower K_D values; 10⁻⁷ to 10⁻⁸M in **Fig. S6**) than factor H (~ mid-10⁻⁷ M (5)) may replace the FH in binding to CspZ. These possibilities have been included in the discussion section (**lines 362 to 363**).

Point-by-Point Response to Reviewers' Comments

Comment 8:

7. Lines 361. Not all mammals have a core temperature of 37 degrees Celsius (close but some are warmer). Also, the term “stays consistent” should be reworded as a “normal core temperature”.

Response:

We rewrote this sentence as “the normal core temperature of mammals stays close to 37°C” (**line 407**).

Comment 9:

8. Lines 380-383. The term “constant immunization” in refer to Ospa vaccination is not accurate. Instead, repeated vaccination is required to maintain a high titered antibody response for efficacy.

Response:

That sentence has been removed/rewritten as suggested by reviewer 2 (see **the response to comment 15 of reviewer 2**). We rewrote the other sentence in the introduction section that has a similar issue (**line 110**).

Comment 10:

9. Lines 387-388. If the premise that the absence of function shown here for CspZ is critical for new epitopes being exposed and protective antibodies generated, then this statement is not well supported as one would need to know binding partners to effectively develop such a pipeline. Deletion of this sentence is suggested.

Response:

We have deleted this sentence, as also suggested by reviewer 1 (see **comment 50 to reviewer 1**).

Comment 11:

Minor Issues:

1. Line 98. The end of this sentence should be reworded for clarity.

Response:

That sentence has been restructured for clarity (**lines 97 to 98**).

Comment 12:

2. Lines 180-182. This sentence needs to be fixed. There seem to be some words missing.

Response:

We rewrote this sentence (**lines 190 to 191**).

Comment 13:

3. Lines 182-188. Please define what the term BA50 is referring to. Individuals who do not do this type of analysis may not know this terminology.

Point-by-Point Response to Reviewers' Comments

Response:

Please refer to comment 6 for this reviewer.

Comment 14:

4. Lines 190-191. The term “doses” is used here. Aren't these immunizations?

Response:

This should be immunizations, instead of doses and it has been corrected (**line 203**).

Comment 15:

5. Fig. 2C and 2E. I understand that it might be busier, but the x-axis should be labeled for these panels. This same issue is pertinent to Fig. 3 (B and C) and Fig. 6 (C and D) as well.

Response:

We have added labels to Fig. 2C, 2E, and 8B. However, the data from Fig. S3 has now incorporated into Fig. 3A-F and M-R (see **comment 8 to reviewer 2**). As there may not be sufficient space to include the x-axis labels of Fig. 3 A to C, H to I, and M to O, we thus did not include those labels in Fig. 3 (**Fig. 2, 3, and 6**).

Comment 16:

6. :Lines 192-195. This sentence should be re-written for conciseness.

Response:

This has been re-written (**lines 205 to 206**).

Comment 17:

7. Lines 202-204. This seems implicit and can be re-written to reflect the uninfected control data.

Response:

This has been re-written (**line 213 to 215**).

Comment 18:

8. Fig. 4. CspZ-YA-C183S is not observed. It this because it is completely superimposable to CspZ-YA?

Response:

We have used the colors that are less mutated so the CspZ-YA-C183S can be observed. (**Fig. 5; see comment 26 to reviewer 1**).

Comment 19:

9. Lines 263-265 and Fig. 5C. F210 is not visible in Fig. 5C. It is observed in Fig. 5D.

Point-by-Point Response to Reviewers' Comments

Response:

This sentence has been rewritten as "...a hydrophobic core between helices G, H and I where residues Y207 and Y211 are on one side, along with F176, I183 and F217 on the other side..." (lines 280 to 282) and those amino acids are now shown in Fig. 6C.

Comment 20:

10. Lines 338-340. This sentence is oversold in its description and is awkwardly worded.

Response:

We rewrote this sentence (lines 379 to 380).

s

Comment 21:

11. Lines 355-358. This sentence is awkwardly worded.

Response:

That sentence has been removed.

Comment 22:

12. Line 363 end of sentence. Replace "mammal uses" with "mammals".

Response:

That sentence has been removed.

Comment 23:

13. Line 374 end of sentence. "investigation" instead of "investigations"

Response:

This grammatical error has been corrected (line 404).

Comment 24:

14. Line 376 second word. Delete the first "the" in this line.

Response:

"The" has been deleted.

Comment 25:

15. Line 379 end of sentence. Would replace "protectivity" with "promotes (or allows) protection".

Response:

That word has been removed and the sentence has been rewritten.

Comment 26:

Point-by-Point Response to Reviewers' Comments

16. Line 383. The term “sparked off” is slang. Please modify.

Response:

That word has been removed and the sentence has been rewritten.

Comment 27:

17. Line 384. The word “efficacy” should be “efficacious”.

Response:

“Efficacy” has been replaced by “efficacious” (**line 418**).

Comment 28:

18. Line 385. Would rephrase “concept-proof” as “proof of concept”.

Response:

This has been rephrased as suggested (**line 418**).

Comment 29:

19. Line 387. The term “inefficacious” could be switched to “nonefficacious”.

Response:

This term has been switched as suggested (**line 420**).

Comment 30:

20. Line 406. Where are BALB/c mice referred to here used? For the monoclonal antibodies?

Response:

BALB/c C3-deficient mice were from in-house breeding colonies to generate *B. burgdorferi*-infected nymphal ticks. This information has been added (**line 440 to 441**).

Comment 31:

21. Line 430. The term “protein-derived mutant protein” is not clear.

Response:

The “protein-derived mutant protein” has been replaced by “CspZ and its mutant derivatives” (**line 466**).

Comment 32:

22. Line 585-603. Where are the phagocytosis assays described here?

Response:

These phagocytosis assays were performed in Fig. S6E. For clarity, we have cited that figure panel (**line 669**).

Point-by-Point Response to Reviewers' Comments

References:

1. K. Brangulis *et al.*, CspZ variant-specific interaction with Factor H incorporates a metal site to support Lyme borreliae complement evasion. *The Journal of biological chemistry* 10.1016/j.jbc.2024.108083, 108083 (2024).
2. U. Lundberg *et al.*, Preclinical Evidence for the Protective Capacity of Antibodies Induced by Lyme Vaccine Candidate VLA15 in People. *Open forum infectious diseases* **11**, ofae467 (2024).
3. M. C. Gingerich *et al.*, Intranasal vaccine for Lyme disease provides protection against tick transmitted *Borrelia burgdorferi* beyond one year. *NPJ Vaccines* **9**, 33 (2024).
4. H. D. Kamp *et al.*, Design of a broadly reactive Lyme disease vaccine. *NPJ Vaccines* **5**, 33 (2020).
5. A. L. Marcinkiewicz *et al.*, Structural evolution of an immune evasion determinant shapes pathogen host tropism. *Proceedings of the National Academy of Sciences of the United States of America* **120**, e2301549120 (2023).

Point-by-Point Response to Reviewers' Comments

Reviewer #1:

The revised version of the paper is much improved, also due to the integration of new data analyses carried out on the structures. The authors properly addressed all my comments and included a new main figure (Figure 7), illustrating the MD and epitope predictions.

Comment 1:

Please add in the MLCE reference (Scarabelli et. al., 2010) in Line 263. Since MLCE is based on identifying amino acids pertaining to flexible hot spots, the N- and C-termini are often picked out erroneously as epitopes, since the termini of a protein are often “naturally” flexible. Maybe a phrase should be added to mention this.

Response:

The MLCE reference has been added and the text has been updated according to the suggestions as well (**line 262 to 268**).

Comment 2:

The English has been improved but I still find the discussion too long (it is longer than the introduction) and it lacks structure. For example, Lines 348 -351 describe what structural vaccinology can do. Then the subject changes and then in Line 368, we go back to what structural vaccinology can be used for. There are many results that are directly repeated rather than summarising them in the context of the field/state-of-the-art.

Response:

The indicated discussion section has been revised according to the suggestion for better flow and clarity (**line 347 to 368**).

Reviewer #2:

Comment 1:

Line 190 – I am not sure OspC type A is the most prevalence in North America. If so please add a recent reference.

Response:

The reference has been added (**line 191**)

Comment 2:

Line 203 – Mice don't get Lyme disease. It is more accurate to say “from *B. burgdorferi* infection with ..”

Response:

“Lyme disease” has been replaced by “*B. burgdorferi* “ (**line 203**).

Comment 3:

Line 227-230 – given that your OspA vac mice have inflammatory scores similar to the control it is prudent to tone down this link between OspA vaccination and inflammation in the joint.

Response:

Point-by-Point Response to Reviewers' Comments

We rewrote this by only describing the results, toning down the link between OspA vaccination and joint inflammation and describe only the results (**line 226 to 229**).

Comment 4:

The reference list remains too long for an original research article: as this is not a review article you can use 1 reference for generalist statements (ex line 348).

Responses:

The reference list has been updated to concise the number of cited articles.

Reviewer #3:

Comment 1:

Line 87. The term “sparks off” is too conversational. Consider changing this term to “promotes” or something similar.

Response:

“Sparks off” has been replaced by “facilitates the development of “ (**line 87**).

Comment 2:

Lines 124. Please consider altering this sentence to read “...vertebrate hosts and, as such, induce CspZ-specific...”.

Response:

We updated the text according to the suggestion (**line 124**)

Comment 3:

Lines 160-162. This sentence is awkward (particularly the verbiage “rarely being at a distance”). Perhaps this sentiment can be separated into two sentences for clarity. Please consider revising.

Response:

We revised that sentence for clarity (**line 160 to 162**).

Comment 4:

Lines 207. Put “control” in place of “controlled”.

Response:

That has been replaced as suggested (**line 206**).

Comment 5:

Line 361. Suggest replacing “it” with “from”.

Response:

That has been replaced as suggested (**line 358**).

Comment 6:

Point-by-Point Response to Reviewers' Comments

Line 404. The end of the sentence (... , requiring further investigation.”) is redundant to the first part of the sentence and could be omitted.

Response:

We removed the text “requiring further investigation.”